# A mobile endocytic network connects clathrin-independent receptor endocytosis to recycling and promotes T cell activation

Ewoud B. Compeer [1,4], Felix Kraus [1,5], Manuela Ecker[1], Gregory Redpath[1], Mayan Amiezer[1,6], Nils Rother[1,7], Philip R. Nicovich[1], Natasha Kapoor-Kaushik[1], Qiji Deng[1], Guerric P.B. Samson[2], Zhengmin Yang[1], Jieqiong Lou[1], Michael Carnell[3], Haig Vartoukian[1], Katharina Gaus[1] & Jérémie Rossy [1,2]

Endocytosis of surface receptors and their polarized recycling back to the plasma membrane are central to many cellular processes, such as cell migration, cytokinesis, basolateral polarity of epithelial cells and T cell activation. Little is known about the mechanisms that control the organization of recycling endosomes and how they connect to receptor endocytosis. Here, we follow the endocytic journey of the T cell receptor (TCR), from internalization at the plasma membrane to recycling back to the immunological synapse. We show that TCR triggering leads to its rapid uptake through a clathrin-independent pathway. Immediately after internalization, TCR is incorporated into a mobile and long-lived endocytic network demarked by the membrane-organizing proteins flotillins. Although flotillins are not required for TCR internalization, they are necessary for its recycling to the immunological synapse. We further show that flotillins are essential for T cell activation, supporting TCR nanoscale organization and signaling.

[1] EMBL Australia Node in Single Molecule Science, School of Medical Sciences and the ARC Centre of Excellence in Advanced Molecular Imaging, University of New South Wales, High St Gate 9, Sydney, NSW 2052, Australia. [2] Biotechnology Institute Thurgau at the University of Konstanz, 8280 Kreuzlingen, Switzerland. [3] Biomedical Imaging Facility, University of New South Wales, High St Gate 9, Sydney, NSW 2052, Australia. [4] Present address: Kennedy Institute of Rheumatology, University of Oxford, Roosevelt Drive, Headington, Oxford OX3 7FY, UK. [5] Present address: Department of Biochemistry and Molecular Biology, Monash University, 23 Innovation Walk, Melbourne, VIC 3800, Australia. [6] Present address: The Garvan Institute of Medical Research, 384 Victoria St, Darlinghurst, NSW 2010, Australia. [7] Present address: Department of Nephrology, Radboud University Medical Center, Geert Grooteplein 10, 6525 GA, Nijmegen, The Netherlands. These authors contributed equally: Ewoud B. Compeer, Felix Kraus. Correspondence and requests for materials should be addressed to J.R. (email: jeremie.rossy@bitg.ch)

The plasma membrane is a highly dynamic environment, which constantly exchanges lipids and proteins with intracellular compartments through exocytic and endocytic processes. Central to the two-way relationship between the plasma membrane and intracellular compartments is endocytic recycling[1]. Recycling returns endocytosed receptors to the plasma membrane and by doing so controls their level of surface expression and consequently the sensitivity of the cell to extracellular stimuli. Many cellular processes such as cytokinesis, transcytosis, morphogenesis, or synaptic transmission rely on recycling[2]. Targeted endocytic recycling to functionally distinct areas of the plasma membrane is one of the main mechanisms through which polarized cells generate and maintain a spatially distinct distribution of membrane proteins[2,3]. Polarized recycling is especially critical for cell migration[4], cell cytokinesis[5], the basolateral polarity of epithelial cells[3], and T cell activation[6,7]. However, little is known about recycling endosome structure, composition, or how they fulfill their function.

In activated T cells, polarized endocytic recycling is the result of a sequence of cellular events starting with kinase-mediated signaling[8] and ending with the translocation of the microtubule-organization center (MTOC) and associated endosomes to the immunological synapse[9]. Endocytic recycling plays a fundamental role in T cell activation[7,10–14], fine-tuning levels of T cell receptor (TCR) and effectors available for signaling, spatially organizing the immunological synapse[15,16] and directly contributing to signaling[17–19]. Despite their essential contribution to T cell activation, cellular mechanisms that coordinate internalization of surface receptors with sustained delivery to the plasma membrane remain incompletely understood. The recycling machinery delivering TCR to the immunological synapse is complex. Several Rab GTPases[6], the intraflagellar transport system protein IFT20[7,20] and sorting nexin 17[21] have been reported to bring TCR back to the cell surface. What unifies these various elements of TCR recycling into a coherent molecular mechanism, and how TCR is sorted for recycling in intracellular compartments is currently unknown.

The membrane organizing protein flotillins have been reported to define a clathrin-independent endocytic route[22,23] and support the recycling of cell surface proteins[24–26]. Here we used a combination of approaches to investigate TCR at each step of its endocytic journey; from the plasma membrane to endosomes and back at the cell surface. We show that in activated T cells TCR is internalized through a clathrin-independent pathway into a mobile and long-lived endocytic network supported by flotillins, which controls its recycling to the immunological synapse. In contrast to clathrin-coated vesicles, which dissociate after cargo delivery to intracellular compartments, flotillins were incorporated at the level of the plasma membrane within the vesicles, building-up the TCR endocytic network. Our results further suggest that the recycling supported by flotillin-positive endosomes provides a critical contribution to T cell activation by regulating the nanoscale organization of TCR at the immunological synapse and promoting phosphorylation of signaling proteins, and the nuclear import of transcription factors.

## Results

**T cell activation promotes T cell receptor complex subunit ζ (TCRζ) but not Lck internalization**. Internalization of T cell receptors and associated signaling proteins has been measured predominantly by flow cytometry, which involves bulk measurements and provides no access to the dynamics of vesicle generation or movement. Here, we used a photoactivation approach to visualize and quantify the internalization of TCRζ, and kinase Lck in resting and activated T cells (Fig. 1a, b). Jurkat T cells expressing TCRζ or Lck fused to a photoactivatable mCherry (PA-mCherry) were deposited on non-activating (poly-L-lysine) or activating (antibodies against CD3ε and CD28) cover glasses and imaged between 10 and 40 min after initial surface contact on a confocal microscope at 37 °C. Restricted areas of the plasma membrane were briefly illuminated with 405 nm light to trigger localized photoactivation of PA-mCherry. Internalization of photoactivated TCRζ or Lck was measured using a custom-made analysis routine that quantifies the number of vesicles detected in each frame of the time series. This approach showed that TCRζ underwent constitutive internalization in resting cells (Fig. 1a, c and Supplementary Video 1), in accordance with previous observations[13,27]. Endocytosis of Lck was less pronounced than that of TCRζ. In activated cells, we observed that TCRζ was rapidly internalized, with most initially photoactivated molecules incorporated into vesicles within 20 s after photoactivation (Fig. 1b, c, and Supplementary Video 2). By contrast, photoactivated Lck molecules were strictly retained at the cell surface in activated cells (Fig. 1b, c and Supplementary Video 3).

We next investigated how the rapid endocytosis of TCRζ was connected to TCR signaling, using Jurkat T cell lines lacking essential components of early TCR signaling; the kinases Lck (JCaM1), and Zap70 (P116) or the adaptor protein Lat (knock-out by CRISPR/Cas9 gene editing). In the absence of Lck or Zap70, TCRζ was internalized significantly less upon TCR activation than in wild-type cells, forming a similar number of vesicles as in resting cells (Fig. 1d, Supplementary Fig. 1a). However, knocking-out Lat, which is directly downstream of Zap70 in the TCR signaling cascade, did not impair TCRζ endocytosis (Fig. 1d). Altogether, these results demonstrate that kinase-mediated early signaling events immediately downstream of TCR activation promote fast and selective endocytosis of TCRζ.

**TCRζ complete endocytosis requires dynamin and actin**. We next investigated the mechanisms responsible for scission of TCRζ containing vesicles from the plasma membrane. Cells expressing TCRζ-PA-mCherry were activated as in Fig. 1 and treated with 80 μM of the dynamin inhibitor dynasore[28] or 100 μM of the actin branching proteins Arp2/3 inhibitor CK666[29]. Neither treatment affected the number of vesicles detected with the analysis routine (Supplementary Fig. 2a, b). However, the punctate TCRζ-PA-mCherry structures identified as vesicles were completely static—as demonstrated by the quantification of their track length (Fig. 1e, f)—and showed increasing fluorescence intensity during the time series, even when located outside of the area initially illuminated with 405 nm light (Fig. 1g, h). This can be explained if these vesicles were still connected to the plasma membrane in the dynasore/CK666-treated cells, enabling them to accumulate photoactivated molecules of TCRζ-PA-mCherry coming from the photoactivated region of the cell. Together with the fact that flotillin vesicles have been shown to rely on microtubules rather than actin for their mobility[26,30], these observations suggest that the immobility of TCRζ-PA-mCherry vesicles upon inhibition of dynamin activity and actin branching results from incomplete scission from the plasma membrane.

**TCRζ endocytosis is clathrin independent**. There is conflicting evidence as to whether TCR is internalized through a clathrin-dependent[31,32] or clathrin-independent[11] route. Thus, we used the clathrin inhibitor pitstop2[33] to investigate the role of clathrin in the rapid, activation-dependent endocytosis of TCRζ. 6 μM pitstop2 drastically reduced the number of vesicles defined by clathrin-EGFP (Supplementary Fig. 2c) and prevented internalization of transferrin (Supplementary Fig. 2d), but did not

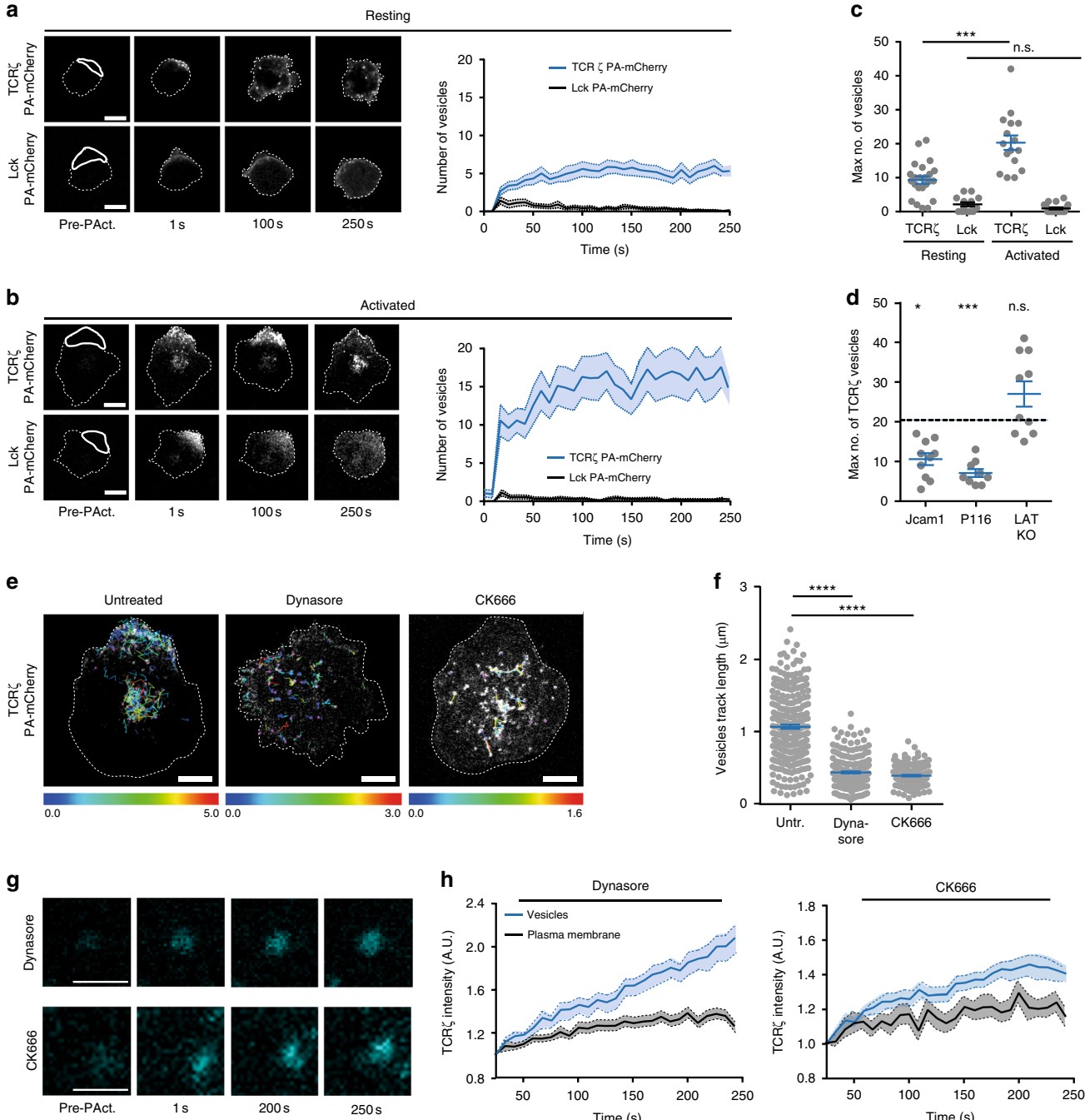

**Fig. 1** T cell activation and signaling promote fast internalization of TCRζ but not of Lck. **a** Left: Representative images of Jurkat T cells expressing TCRζ-PA-mCherry (top panels) or Lck-PA-mCherry (bottom panels), allowed to adhere on non-activating (poly-L-lysine) or **b** activating (anti-CD3ε and anti-CD28) surfaces, photoactivated on membrane region of interest and subsequently imaged for 250 s. Images show T cells before and after activation and at indicated time points. Right: number of PA-mCherry vesicles detected in each frame during the time of acquisition. **c** Maximum number of PA-mCherry vesicles detected in a given frame during the 250 s acquisition time. Each dot represents a cell. **d** Maximum number of PA-mCherry vesicles detected in Jurkat cells lacking functional Lck (JCam1), Zap70 (P116) or Lat (LAT KO) and activated on anti-CD3ε and anti-CD28 coated surfaces. Horizontal dashed line represents the values for TCRζ in activated cells from panel c. **e** Representative examples of TCRζ-PA-mCherry vesicle tracks detected in activated untreated cells or treated with dynasore or CK666. Color scale is length in μm. **f** Length of vesicle tracks in untreated, dynasore or CK666 treated cells. Each dot is one track. **g** Zoomed images of immobile TCRζ-PA-mCherry vesicles in dynasore (top) or CK666 (bottom) treated Jurkat T cells. Scale bar is 0.5 μm. **h** Fluorescence intensity profiles over time of single vesicles, such as those shown in **g** and neighboring plasma membrane regions of the same size. Scale bars, 5 μm. Data obtained from three or more independent experiments. Small horizontal lines indicate mean (±SEM). ns, not significant; *$p < 0.01$; ***$p < 0.0001$; ****$p < 0.00001$; Mann–Whitney $t$-test

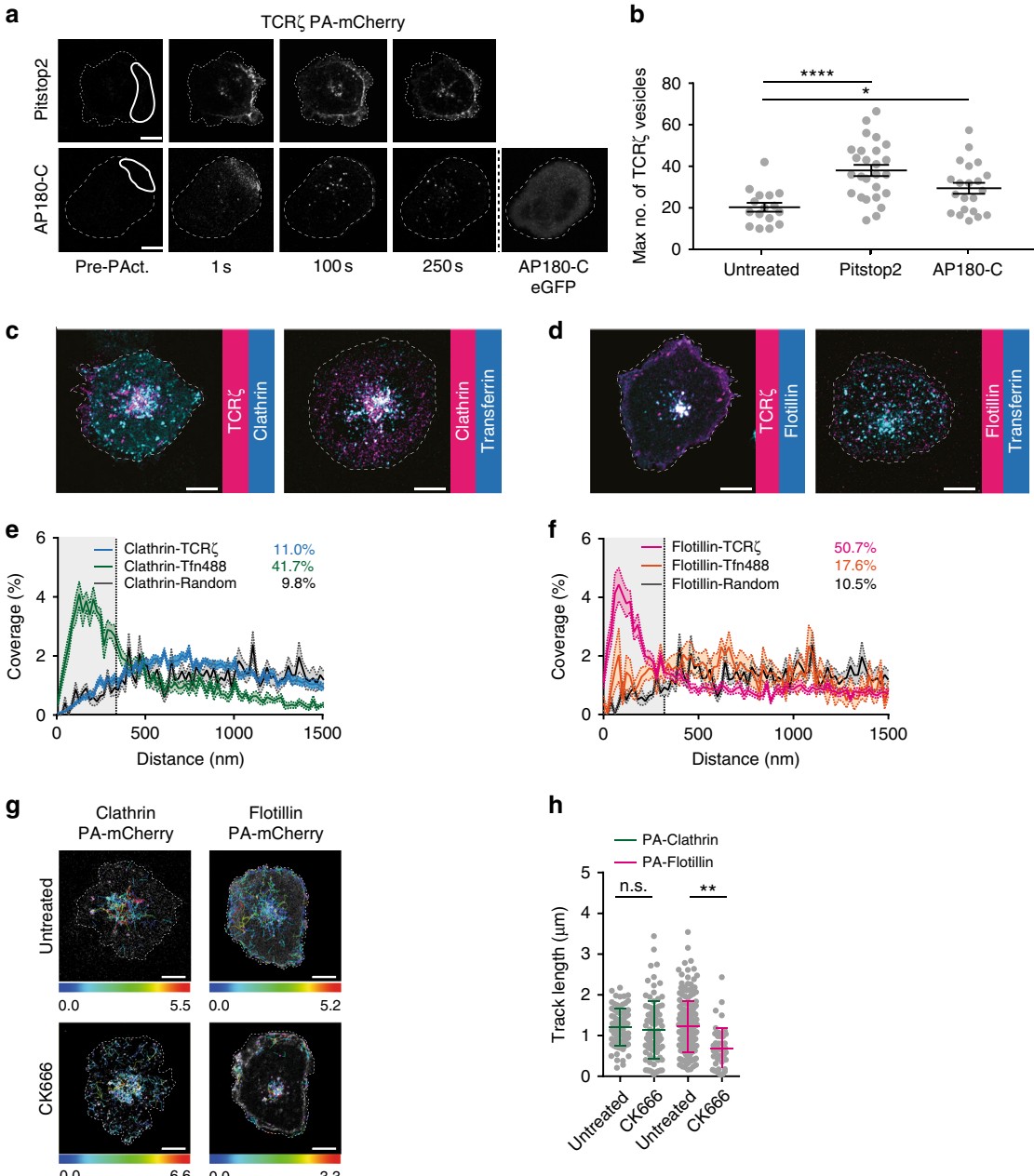

**Fig. 2** TCRζ is internalized through a clathrin-independent pathway in vesicles positive for flotillin-2. **a** Representative images of activated Jurkat T cells expressing TCRζ-PA-mCherry either pretreated with 6 μM pitstop2 or co-expressing AP180-C-EGFP and activated and photoactivated as before. **b** Maximum number of PA-mCherry vesicles in cells treated with pitstop2 or co-expressing AP180-C-EGFP. **c**–**f** Representative images (**c** and **d**) and cross-channel nearest neighbor distance (**e** and **f**) between vesicles defined by Clathrin-EGFP and TCRζ-PA-mCherry or Clathrin-PA-mCherry and transferrin-Alexa488, and flotillin-2-EGFP and TCRζ-PA-mCherry or flotillin-2-PA-mCherry and transferrin-Alexa488, in Jurkat T cells activated on anti-CD3ε and anti-CD28-coated surfaces. Percentages shown are for vesicles having a distance of less than 320 nm to their nearest neighbor in the other channel. **g** Representative examples of Clathrin-PA-mCherry (left) or flotillin-2-PA-mCherry (right) vesicle tracks detected in activated T cells untreated (top) or treated with CK666 (bottom). Color scale depicts length in μm. **h** Length of vesicle tracks in vehicle-treated and CK666-treated cells. Each dot represents a single track. Scale bars, 5 μm. Data obtained from three or more independent experiments. Small horizontal lines indicate mean (±SEM). ns, not significant; **p < 0.001; ****p < 0.00001; Mann–Whitney t-test

decrease the rapid internalization of TCRζ upon TCR activation (Fig. 2a, b). On the contrary, clathrin inhibition increased the number of TCRζ endocytic vesicles, suggesting a cross-regulation between clathrin-mediated endocytosis and an alternative clathrin-independent endocytic route[34]. We further tested the clathrin dependence of TCRζ internalization in cells co-expressing TCRζ-PA-mCherry and the c-terminal domain of AP180 (AP180-C), which blocks clathrin-coated pit formation[35].

AP180-C did not prevent TCRζ internalization (Fig. 2a, b), further indicating that TCRζ endocytosis does not require the formation of clathrin-coated pits in activated T cells.

To confirm these findings, we assessed to what extent clathrin-coated vesicles coincided with endocytosed TCRζ. Cells were co-transfected with clathrin fused to EGFP and TCRζ fused to PA-mCherry, deposited on activating surfaces and photoactivated as in Fig. 1. For each frame of the times series, we calculated the

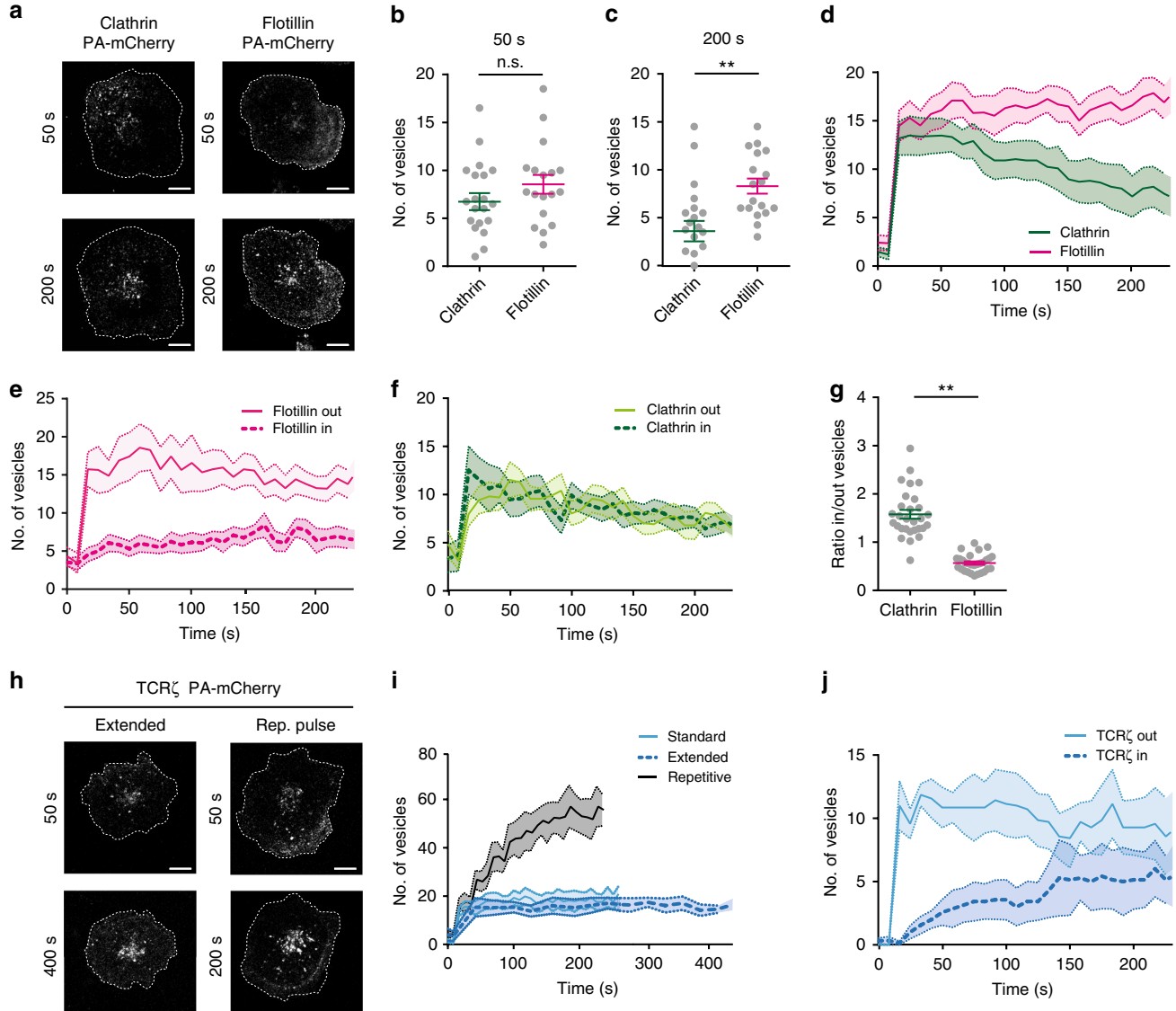

**Fig. 3** Internalized flotillins and TCRζ define a long-lived mobile endocytic network. **a** Representative images of Jurkat T cells expressing Clathrin-PA-mCherry (left) or Flotillin-PA-mCherry (right), activated on anti-CD3ε and anti-CD28-coated surfaces, photoactivated on a region of interest and subsequently imaged for 250 s. Images are from 50 and 200 s after photoactivation. Number of PA-mCherry vesicles detected in each frame at (**b**) 50 s and (**c**) 200 s. **d** Number of PA-mCherry vesicles detected at any given point during 250 s acquisition. **e** Number of PA-mCherry vesicles detected within or outside of the central endosomal region for each frame during the time of acquisition for Flotillin2-PA-mCherry and (**f**) Clathrin-PA-mCherry. **g** Ratio of Flotillin2-PA-mCherry or Clathrin-PA-mCherry positive vesicles detected in or out of the central endosomal region. **h** Example of Jurkat T cells expressing TCRζ-PA-mCherry after photoactivation of a region of interest and imaged for 450 s (extended) or in cells repetitively photoactivated every 8 s (repetitive). **i** Number of PA-mCherry vesicles detected in each frame during the time of acquisition as in Fig. 1b (standard) or during extended and repetitive acquisition. **j** Number of TCRζ-PA-mCherry vesicles detected within or outside of the central endosomal region for each frame during the time of acquisition. Each dot represents a cell. Scale bars, 5 μm. Data obtained from three or more independent experiments. Small horizontal lines indicate mean (±SEM). ns, not significant; **p < 0.001, Mann–Whitney t-test

cross-channel nearest neighbor distance between vesicles in the green channel (clathrin-EGFP) and vesicles identified in the red channel (TCRζ-PA-mCherry). Taking into account the average size of vesicles and the resolution of confocal microscopy, a vesicle in the first channel was considered to be the same vesicle as that in the second channel if its nearest neighbor in the second channel was within 320 nm. In accordance with the results obtained with pitstop2, endocytic vesicles defined by TCRζ-PA-mCherry did not coincide with vesicles defined by clathrin-EGFP more than with vesicles randomly distributed throughout the cell (11%, Fig. 2c). Importantly, vesicles defined by clathrin-EGFP did overlap in space and time with the canonical clathrin cargo

transferrin, labelled with Alexa488, significantly more than they did with randomly distributed vesicles (41.7%, Fig. 2c).

**TCRζ is internalized in vesicles positive for flotillin-2.** In order to further characterize the clathrin-independent endocytic route mediating TCR internalization, we investigated the membrane-organizing proteins flotillin-1 and 2 that are known to support clathrin-independent endocytosis[22,23]. In contrast to what was measured for clathrin, 50.7% of endocytic vesicles of TCRζ-PA-mCherry coincided with vesicles defined by flotillin2-EGFP in space and time (Fig. 2d). Of note, flotillin 1 and 2 form hetero-

oligomer in vesicles[36] and we obtained an almost identical score, 54.6%, of flotillin2-PA-mCherry vesicles less than 320 nm to the nearest flotillin1-EGFP containing vesicles (Supplementary Fig. 3). Flotillin-2-PA-mCherry vesicles did not correlate more with transferrin-Alexa488 positive vesicles than randomly distributed vesicles (Fig. 2d). In line with these results, flotillin-2 positive vesicles shared the same actin requirement as TCRζ for fission from the plasma membrane; Arp2/3 inhibition by CK666

did not affect the movement of clathrin-coated vesicles but resulted in immobilized punctate structures defined by flotillin2-EGFP (Fig. 2e, f). Interestingly, pitstop2 had the same positive regulation on vesicles defined by flotillin-2 (Supplementary Fig. 2c) than it had on TCRζ endocytic vesicles (Fig. 2b). Taken together, these data show that TCRζ proteins are not internalized via a clathrin-mediated pathway in activated T cells, but rather follow an endocytic route defined by flotillins that relies on the

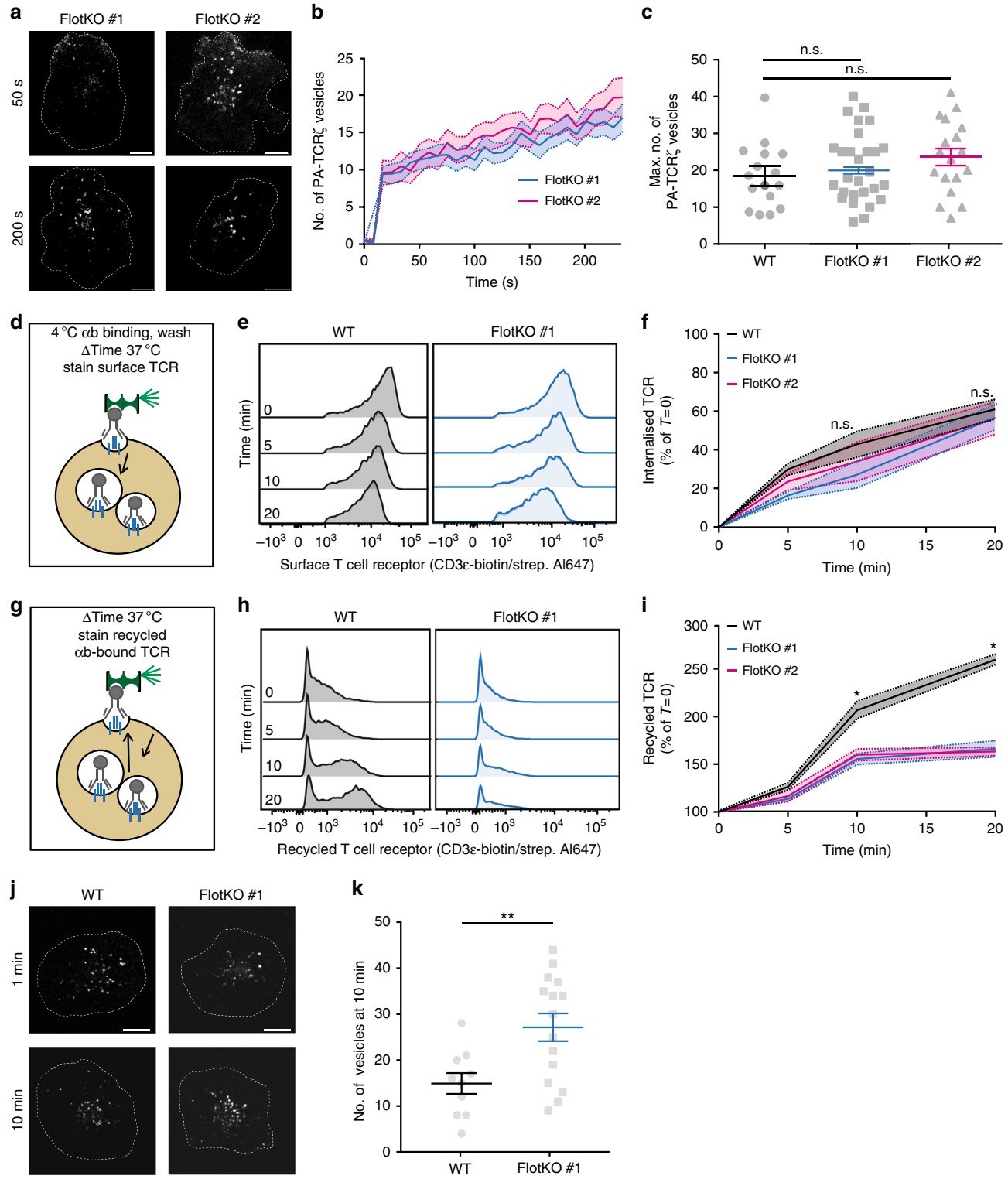

activity of dynamin and Arp2/3 for endosome fission from the plasma membrane.

### TCRζ is internalized into a flotillin-positive endocytic network.

To gain a better insight into the endocytic pathway demarked by flotillins in T cells, we compared internalization of clathrin-PA-mCherry and flotillin2-PA-mCherry in activated T cells. Shortly after photoactivation, we detected a similar number of vesicles for clathrin and flotillin (Fig. 3a, b). The number of clathrin-PA-mCherry vesicles decreased with time (Fig. 3c, d), probably due to the dissociation of the clathrin coat[37]. By contrast, the number of flotillin2-PAmCherrry vesicles remained stable for at least 250 s following photoactivation (Fig. 3d). In activated T cells, vesicles containing TCRζ show a radial movement to and from a central endosomal structure visible at the center of the immunological synapse, whose location coincides with the MTOC[16] (Supplementary Fig. 4a). Vesicles defined by flotillin2-PA-mCherry moved continuously between the periphery (out) and the center (in) of the immunological synapse (Fig. 3e, g) and did not accumulate in the central endosomal region that was identified by its high fluorescence intensity (Fig. 2e, Supplementary Fig. 4b). Consistent with coat dissociation, clathrin-coated vesicles did not display the same dynamics, as their number decreased equally over time at the center and periphery of the cell (Fig. 3f, g). Similar to vesicles defined by flotillin2-PA-mCherry, the number of vesicles containing photoactivated TCRζ remained stable up to 250 s after photoactivation (Fig. 3i) and moved continuously between the periphery and center of the immunological synapse (Fig. 3j). The number of TCRζ-PA-mCherry containing vesicles remained stable for even up to 450 s after the initial photo-activation (Fig. 3h). Repetitive photoactivation of TCRζ-PA-mCherry at the same plasma membrane area every 8 s led to a constant increase in the number of TCRζ-PA-mCherry vesicles (Fig. 3h, i). This suggests that TCRζ-PA-mCherry molecules diffused fast enough to at least partially replete the photoactivated region of interest within the 8 s between the photoactivation pulses. The steady number of vesicles for 450 s following one activation and the increasing number of vesicles resulting from repetitive photoactivation indicate that, following internalization, TCRζ is not returned to the plasma membrane and remains in endocytic compartments for a significant amount of time. Alto-gether, these data suggest that flotillin-positive vesicles define a mobile endocytic network that contributes to TCRζ intracellular trafficking. Importantly, flotillins remain in the vesicles con-stituting this network after endocytosis, in contrast to clathrin that is not incorporated into the intracellular compartments to which it delivers cargo[37].

### Flotillins are required to recycle but not to endocytose TCR.

We set out to determine the precise contribution of the endocytic network defined by flotillins in TCRζ endocytic trafficking. We generated a Jurkat T cell line in which both flotillin-1 and flotillin-2 were knocked-out using CRISPR/Cas9 gene editing (Supplementary Fig. 5a). Surprisingly, photoactivation experiments did not reveal any difference in the dynamics or extent of the formation of TCRζ-PA-mCherry endocytic vesicles in flotillin1/2 KO T cells compared to WT T cells (Fig. 4a–c, Supplementary Video 4), suggesting that flotillins do not con-tribute to TCR endocytosis. Flotillin knock-out had no observable effect either on the clathrin-mediated transferrin internalization (Supplementary Fig. 5b-d). The flotillin-independence of TCR endocytosis was further confirmed using a flow cytometry-based internalization assay. TCR complexes present at the cell surface were labeled with activating anti-CD3ε biotinylated antibody in the presence of co-stimulatory anti-CD28 antibody at 4 °C to prevent endocytosis. T cells were then incubated at 37 °C for 0, 5, 10, or 20 min and the amount of non-internalized cell-surface TCR was detected with fluorescent streptavidin (Fig. 4d and Supplementary Fig. 6). In line with the data obtained using photoactivation, flow cytometry revealed no difference in TCR internalization between flotillin1/2 KO and WT T cells (Fig. 4e, f).

While our data clearly showed that TCRζ is internalized in an endocytic network marked by flotillins, flotillin1/2 KO T cells did not show any impairment in TCR endocytosis. Thus, in order to identify the function of flotillin-positive endosomes in intracel-lular trafficking of TCR, we investigated how knocking-out flotillins impacted TCR endocytic recycling. To do so, we measured recycling of TCR complexes back to the plasma membrane using a flow cytometry-based antibody feeding assay[14]. T cells were incubated with activating biotinylated antibody against CD3ε in the presence of co-stimulatory anti-CD28 antibody at 37 °C for 40 min to allow TCR internalization. Antibodies bound to the remaining TCR at the cell surface were then blocked with unlabeled streptavidin. Recycled TCR com-plexes were detected at different time points after the surface block using fluorescently labeled streptavidin (Fig. 4g). T cells lacking flotillins showed a significant reduction in the number of TCR complexes recycled to the plasma membrane from 10 min after the beginning of the recycling in comparison with WT T cells (Fig. 4h, i). To determine if the defect of recycling in the flotillin1/2 KO T cells would lead to an accumulation of endosomes containing TCRζ, we measured the number of TCRζ-PA-mCherry positive vesicles 10 min after photoactivation. At this time point, Flotillin1/2 KO T cells had a higher count of TCRζ-PA-mCherry positive vesicles than the WT cells (Fig. 4j, k). This finding suggests that TCRζ accumulated in intracellular compartments because of the recycling defect in cells lacking flotillins. These results demonstrate that although flotillins do not mediate TCR internalization, they are essential to TCR endocytic recycling during T cell activation.

### Flotillins spatially organize TCRζ endosomes in activated T cells.

In order to understand the role played by flotillin-mediated recycling in delivery of TCR to the immunological synapse upon activation, Jurkat T cells expressing TCRζ-mCherry

---

**Fig. 4** Flotillin expression is required for recycling of TCR but not for its internalization. **a** Flotillin knock-out Jurkat T cells expressing TCRζ-PA-mCherry, activated on anti-CD3ε-coated and anti-CD28-coated surfaces, photoactivated on outer membrane region of interest, and subsequently imaged for 250 s. **b** Number of PA-mCherry vesicles detected for each frame during the time of acquisition. **c** Maximum number of PA-mCherry vesicles detected in a given frame for Flotillin1/2 KO T cells and WT T cells (dotted line). **d** Diagram of internalization flow cytometry assay. **e** Mean fluorescent intensity (MFI) at the surface of 50,000 Jurkat T cells (Left: wildtype; Right: Flotillin1/2 KO) stained with an activating antibody against CD3ε and allowed to internalize TCR complexes for 0, 5, 10, and 20 min. **f** Decrease of MFI at indicated time points relative to $t = 0$. **g** Diagram of antibody feeding assay. **h** MFI of 50,000 Jurkat T cells (Left: wildtype; Right: Flotillin-1/-2 KO) allowed to recycle TCR for 0, 5, 10, and 20 min after having internalizing TCR complexes labelled with an activating antibody against CD30ε for 40 min. **i** MFI increase at indicated time points relative to $t = 0$. **j** Images of WT and flotillin1/2 KO cells on activated surfaces and expressing TCRζ-PA-mCherry at 1 and 10 min after photoactivation. **k** Number of PA-mCherry vesicles detected at 10 min after photoactivation. Scale bar, 5 μm. Data obtained from three or more independent experiments, with at least five cells per experiment. Small horizontal dotted lines indicate mean (±SEM). ns, not significant; *$p < 0.01$, Mann–Whitney $t$-test

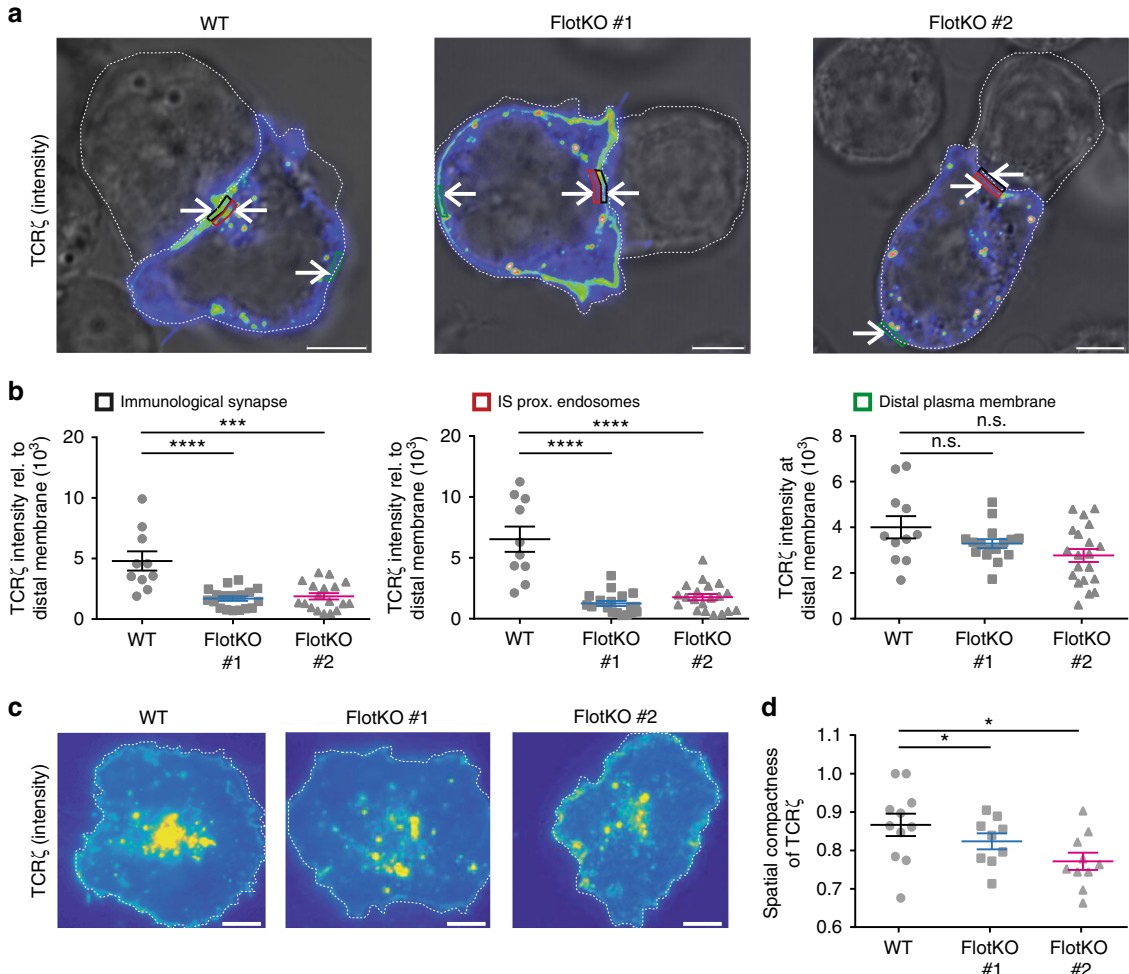

**Fig. 5** Flotillins organize the subcellular localization of TCRζ. **a** Representative images of wildtype and Flotillin1/2 KO Jurkat T cells expressing TCRζ-mCherry and conjugated to SEE pulsed Raji B cells. **b** TCRζ-mCherry intensity at the immunological synapse, a region directly behind the synapse, and at a region in the plasma membrane distal to the immunological synapse. Regions are indicated by corresponding colored boxes in **a** (arrows). **c** Representative TIRF images of wildtype and Flotillin1/2 KO Jurkat T cells expressing TCRζ-mCherry on activating anti-CD3ε and anti-CD28 antibodies coated glass surfaces. **d** Quantification of the 2D spatial distribution of TCRζ-mCherry intensity. A score of 1 means that the total mCherry intensity detected in the image is concentrated within one single pixel. Scale bars, 5 µm. Data obtained from three independent experiments. Error bars indicate mean (±SEM). ns, not significant; *$p < 0.01$; ***$p < 0.0001$; ****$p < 0.00001$; Mann–Whitney $t$-test

were activated for 10 min with Raji B-cells pulsed with super-antigen staphylococcal enterotoxin E (SEE) and fixed. To quantify TCRζ subcellular localization, we measured mCherry fluorescent intensity in three different regions of the cells: (a) at the immunological synapse, (b) directly behind the synapse, and (c) in the plasma membrane distal to the immunological synapse (Fig. 5a). Intensities at the immunological synapse and endosomes were normalized to the intensity measured at the distal region to correct for different expression levels. We detected five times less TCRζ at the immunological synapse and endosomes in Flotillin1/2 KO than in WT T cells (Fig. 5b), illustrating that flotillins are required for delivery and recruitment of TCR to the immunological synapse. We further used total internal reflection fluorescence microscopy (TIRF) to quantify the spatial organization of TCRζ subsynaptic endosomes in cells expressing TCRζ-mCherry and fixed after 10 min of activation on anti-CD3ε-coated and anti-CD28-coated glass surfaces. To assess the distribution of TCRζ-mCherry signal, we developed a measure that reflected the signal compaction relative to the cell area. The most concentrated the signal could theoretically be if it resided within a single pixel (intensity compactness = 1) in contrast to

being equally distributed across all pixels within the cell (intensity compactness = 0). Cells lacking flotillins showed significantly more dispersed TCRζ endosomes than WT cells (Fig. 5c, d). Together with the data obtained with Jurkat–Raji B-cell conjugates, this result demonstrates that flotillins play a determinant role in the spatial organization of TCRζ endosomes in activated T cells.

**Flotillin controls TCRζ cell surface nanoscale organization.** We next used single molecule localization microscopy to investigate how the absence of flotillin could affect the spatial organization of TCR at the plasma membrane. Jurkat WT and flotillin1/2 KO T cells expressing TCRζ fused to the photoswitchable fluorescent protein PS-CFP2 were fixed with paraformaldehyde 10 min after contact with non-activating (antibody against CD90)[38] and activating (antibodies against CD3ε and CD28) coated surfaces[39,40]. Using a DBSCAN approach to identify clusters[39,41], we found that flotillin1/2 KO T cells had a significantly lower number of TCRζ clusters per µm$^2$ at the area of activation than their wildtype counterpart (Fig. 6a, b). We detected two distinct

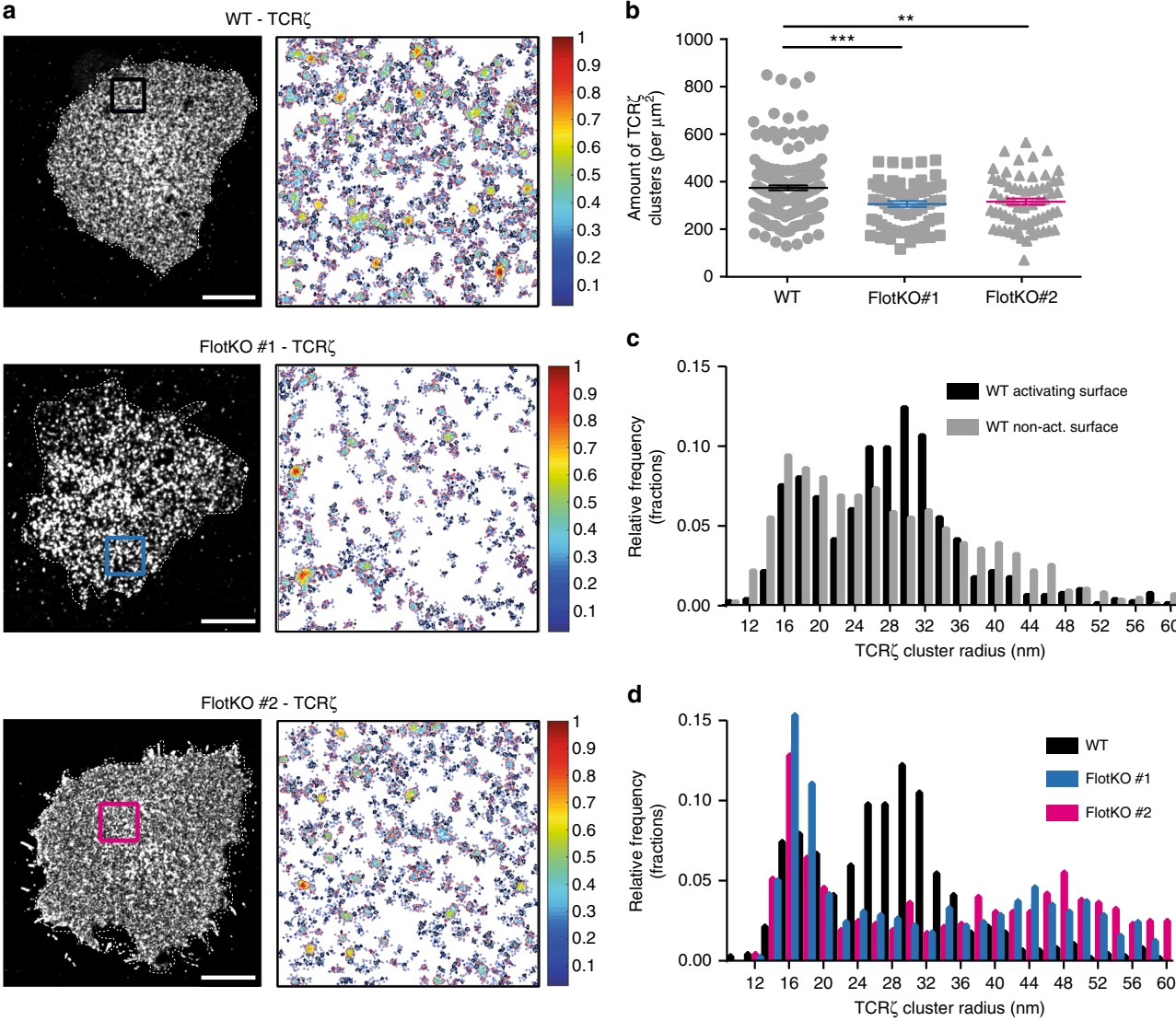

**Fig. 6** Flotillin-mediated TCR recycling controls TCRζ cell surface nanoscale organization. **a** Left: Representative single-molecule images of TCRζ-PSCFP2 in WT (top) and flotillin1/2 KO (bottom) Jurkat T cells on activating anti-CD3ε and anti-CD28 antibodies coated glass surfaces. Right: cluster maps color-coded for the relative molecular density from regions (3 μm × 3 μm) highlighted in the single-molecule images (boxes); normalized relative density is pseudocoloured. **b** Number of TCRζ-PSCFP2 clusters per μm² identified by DB-SCAN, in WT and flotillin1/2 KO Jurkat T cells. **c** TCRζ-PSCFP2 cluster size distribution in WT Jurkat T cells on non-activating (anti-CD90 antibodies; grey) and activating (anti-CD3ε and anti-CD28 antibodies; black) coated glass surfaces. **d** TCRζ-PSCFP2 cluster size distribution in WT and flotillin1/2 KO Jurkat T cells on activating (anti-CD3ε and anti-CD28 antibodies) coated glass surfaces. Scale bars, 5 μm. Data obtained from three or more independent experiments, with at least five cells per experiment. Error bars indicate mean (±SEM). ns, not significant; **$p < 0.001$; ***$p < 0.0001$; Mann–Whitney $t$-test

populations of TCRζ clusters in WT T cells, with an average diameter of $16 \pm 2$ and $30 \pm 4$ nm, respectively, T cell activation promoting the prevalence of the larger $30 \pm 4$ nm cluster population (Fig. 6c). The $30 \pm 4$ nm population—promoted by T cell activation—was completely absent from flotillin1/2 KO cells (Fig. 6d). The $16 \pm 2$ nm cluster population seemed to be insensitive to flotillin depletion, suggesting that the larger $30 \pm 4$ nm population does not result from concatenation of the smaller $16 \pm 2$ nm clusters. Pharmacological inhibition of recycling with primaquine[42] led to a similar reduction in the number of TCRζ molecules, cluster number and importantly to the disappearance of the $30 \pm 4$ nm cluster population as observed in flotillin-1/-2 KO cells (Supplementary Fig. 7). Taken together, these results show that flotillins contribute to the spatial organization of TCRζ molecules at the cell surface by promoting a specific activation-related population of TCRζ clusters.

**Flotillins are required for T cell activation.** Beyond the nanoscale organization of TCR at the immunological synapse, TCR polarized endocytic recycling is required at almost every step of T cell activation for cellular mechanisms such as formation of T cell–antigen-presenting cells conjugates[43] and TCR-triggered signaling[14,15]. To determine the contribution of flotillin-mediated TCR recycling in conjugate formation, we stained Jurkat T cells in red using cell viability dye and Raji B-cells in green using CFDA. After 10 min of incubation, we quantified T–B cell conjugates identified as red–green double positive population in flow cytometry[43]. Without superantigen, only 0.5% of T-cells and B-cells formed conjugates. In the presence of superantigen, 32.3% of wildtype T cells but only 15% of flotillin knock-out T cells formed conjugates with B cells (Fig. 7a, b).

We further used flow cytometry to evaluate the impact of the absence of flotillins on key phosphorylation events of TCR

signaling after activation with soluble antibodies against CD3ε and CD28 for 0, 5, 10, and 20 min. Flotillin1/2 KO T cells showed a significant reduction in phosphorylation of TCRζ, Zap70, and extracellular signal-regulated kinase (ERK) upon T cell activation compared to WT cells (Fig. 7c). Immunofluorescence staining of T–B cell conjugates with antibodies against phosphorylated TCRζ and phosphorylated Zap70 revealed the same tendency, suggesting that the conjugates that were formed had impaired signaling

but were at least partially functional (Supplementary Fig. 8). Artificially aggregating flotillin-positive endosomes with the optogenetic tool Cry2Clust as recently done for Rab11[44,45] reduced the percentage of Jurkat T cells positive for phosphorylated TCR in response to TCR activation (Supplementary Fig. 9). This suggests that impaired T cell activation in flotillin1/2 KO T cells is the result of the impaired TCR recycling observed in these cells.

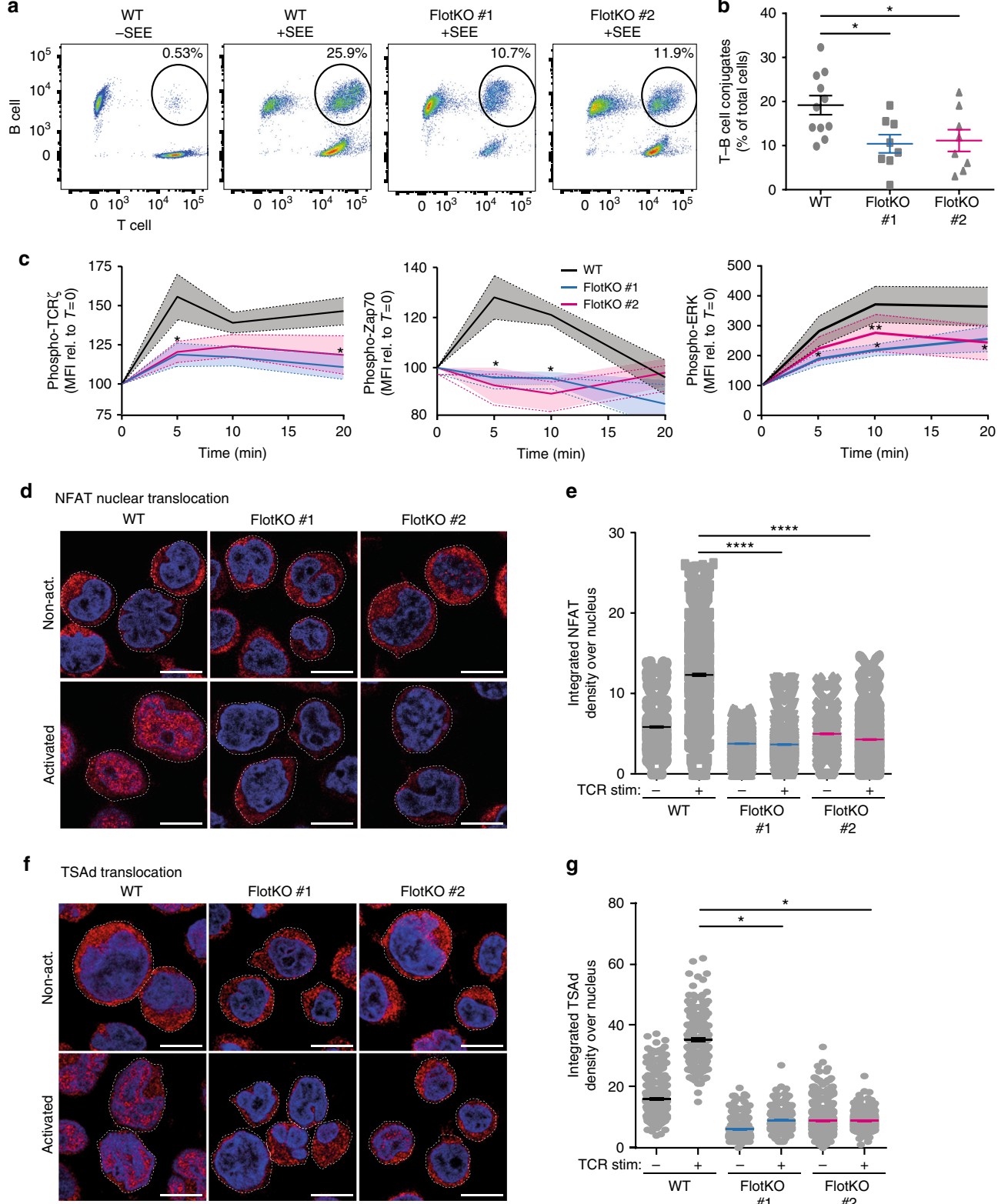

Finally, to understand the role of flotillin-mediated recycling on late TCR signaling events, we used immunofluorescence staining to measure the nuclear localization of nuclear factor of activated T cells (NFAT) and of T cell-specific adaptor protein (TSAd), which both translocate to the nucleus upon T cell activation[46,47]. Quantification of the levels of NFAT or TSAd co-localizing with the nucleus identified by Hoechst staining showed that their nuclear translocation following T cell activation was abolished by flotillin knock-out (Fig. 7d–g).

We confirmed the importance of flotillins for T cell activation in primary CD4 T cells isolated from the spleen of Flotillin2 KO mice[48]. Because splenocytes isolated from these mice had no detectable flotillin1 expression (Fig. 8a), T cells isolated from these mice were de facto double KO. CD4 T cells were stimulated on anti-CD3 and anti-CD28 surfaces for 20 and 36 h and stained with antibodies against CD69 and CD25, which are canonical markers of T cell activation. T cells isolated from flotillin-2 KO mice had significantly less surface expression of CD69 and CD25 after 20 and 36 h, respectively. This demonstrates a role for flotillins in activation of primary T cells. Altogether, our investigation of the functional consequences of flotillins KO suggest that flotillin-mediated TCR endocytic recycling is required for various cellular events critical to T cell activation, from conjugation with antigen-presenting cells to early and late signaling events.

## Discussion

How endocytic recycling operates is incompletely understood, despite its importance in many cellular processes. In this work, we uncovered a novel role for the membrane-organizing flotillins in organizing the recycling of cell surface receptors. We showed that TCR triggering leads to its rapid, clathrin-independent uptake. Immediately after internalization, TCR endocytic vesicles are incorporated into a network of mobile intracellular compartments defined by the membrane-organizing proteins flotillins. We further showed that although flotillins are not required for TCR internalization, they are necessary for recycling TCR to the immunological synapse and for its nanoscale organization at the plasma membrane. More importantly, our results suggest that the recycling supported by flotillin-positive endosomes could be essential to full T cell activation in Jurkat and in primary CD4 T cells. Flotillins are indeed required for conjugation of T cells with antigen-presenting cells, phosphorylation of TCR signaling proteins and the nuclear import of transcription factors, as well as for CD69 and CD25 surface expression in primary CD4 T cells. Collectively, our data support a model in which specialized endocytic sorting machinery underpinned by flotillins promotes the recycling of internalized TCR complexes to the immunological synapse to support T cell activation.

A central aspect of endocytic recycling is the sorting of cargo to membrane domains that eventually bud from recycling endosomes to return to the plasma membrane. Sorting in endosomes relies on soluble tetrameric adaptors, such as AP-1, AP-3 and the Golgi-localizing γ-ear containing ADP ribosylation factor-binding protein (GGA) family[49,50]. These adaptors initiate formation of recycling vesicles by recruiting cargos, directly binding to consensus sequences on the cargo cytosolic domains[51]. While it has been shown that flotillins can influence endosomal sorting through binding to this type of cytosolic consensus sequence[52], TCRζ does not display such a binding motif. Instead of direct binding, TCR and flotillins interact within cholesterol-enriched membrane domains, an association increased upon TCR triggering[53,54]. These interactions could for instance be responsible for the change in TCR clustering at the plasma membrane that we observed in the flotillin1/2 KO T cells. Similarly, concentration of TCR complexes in flotillin-rich membrane domains at the surface of intracellular compartments could support a mechanism to sort them for recycling to the immunological synapse. In the same line, it has recently been shown that cargos internalized through a clathrin-independent route benefit from facilitated sorting in endosomes by remaining compartmentalized within endosomal membrane domains[55]. Our data suggests that such a compartmentalization of TCR throughout internalization and recycling is a key element of T cell activation. In this context, it is possible that the recycling of TCR complexes within flotillin positive vesicles contribute to organize the clustering of TCR at the plasma membrane.

Unlike AP-1 and AP-3 and GGAs, whose subcellular localization is confined to endosomes, flotillins are present at the plasma membrane and in TCRζ endocytic vesicles. Our results suggest flotillins might provide an identification mark to the endocytic cycle of TCR, allowing specific TCR complexes from the plasma membrane to be incorporated into specific recycling compartments. A similar mechanism has been observed in adherent cells where loss of adhesion triggers internalization of lipid raft components and GPI-anchored proteins through clathrin-independent route, while re-adhesion promotes their Arf6-mediated recycling to the plasma membrane[56]. Like our observations on flotillin-mediated recycling, Arf6 is often required for recycling of cargos internalized through a clathrin-independent pathway, but not for their endocytosis[57]. This suggests that the relationship between Arf6 and flotillins in polarized recycling is worth further investigation.

Arf6-associated recycling and recycling of proteins internalized through clathrin-independent pathways are associated with endosomes of specific morphologies and dynamics[57–60]. Our data revealed that the TCR endocytic network delineated by flotillins has a distinct morphology and dynamics, consisting of long-lived vesicles that persistently move behind the immunological

---

**Fig. 7** Flotillin expression is required for conjugate formation and T cell signaling. **a** Representative flow cytometry dot plots of wildtype and Flotillin1/2 KO Jurkat T cells conjugated to SEE pulsed Raji B cells and stained with cell viability dye and CFDA, respectively. **b** Percentage of stable T−B cell conjugates determined as violet dye and CFDA double positives, relative to total cell population. **c** Flow cytometry quantification of WT and flotillin1/2 KO Jurkat T cells activated with soluble CD3ε and CD28 and fixed prior or 5, 10, and 20 min after activation. Cells were stained with antibodies against phosphorylated (left) TCRζ-pY142, (middle) Zap70-pY493, and (right) ERK-pY202/204. **d** Representative immunofluorescence microscopy images of WT and flotillin1/2 KO Jurkat T cells stained with antibodies against nuclear factor of activated T-cells (NFAT) and **e** the integrated NFAT fluorescence intensity over the Hoechst-stained nucleus in WT and flotillin1/2 KO Jurkat T cells activated or not with anti-CD3ε and CD28 antibodies. **f** Representative immunofluorescence microscopy images of WT and flotillin1/2 KO Jurkat T cells stained with antibodies against T-cell-specific adaptor protein (TSAd) and **g** the integrated TSAd fluorescence intensity over the Hoechst-stained nucleus in WT and flotillin1/2 KO Jurkat T cells activated or not with anti-CD3ε and CD28 antibodies. Scale bars, 5 μm. Data obtained from three or more independent experiments. Error bars indicate mean (±SEM). *$p < 0.01$; **$p < 0.001$; ****$p < 0.00001$; Mann–Whitney $t$-test

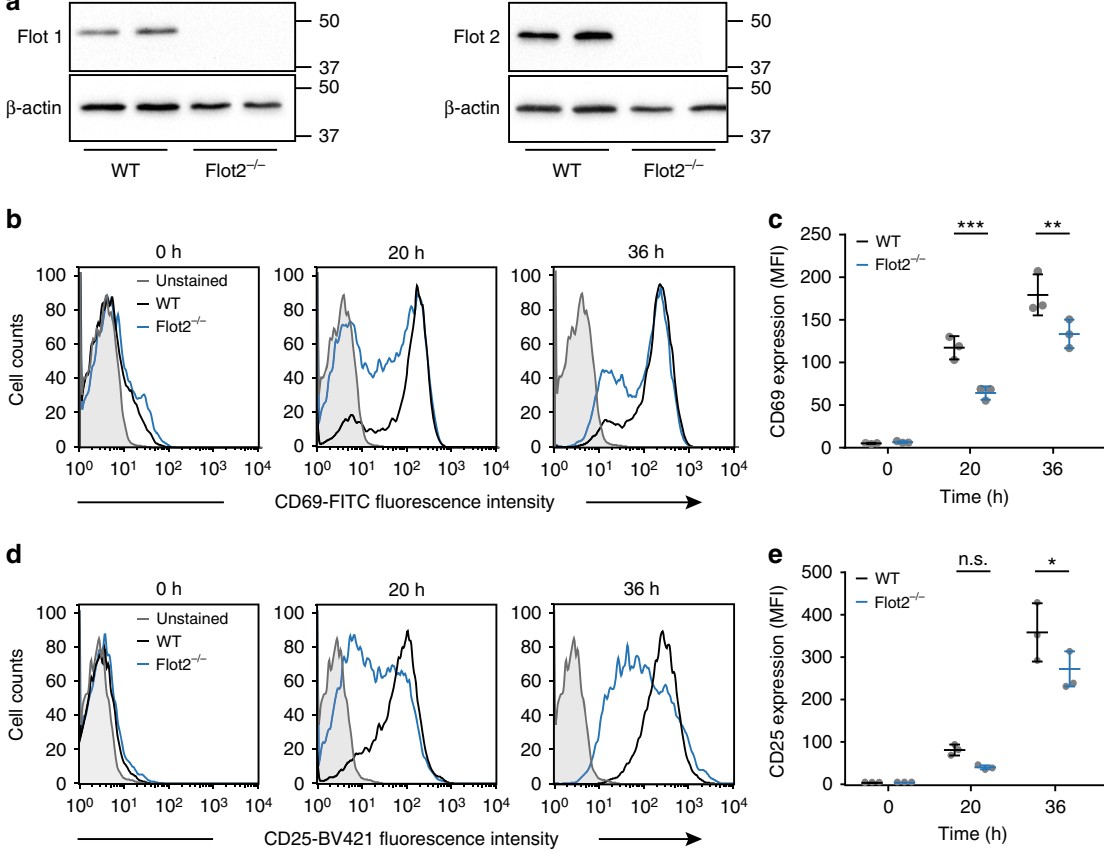

**Fig. 8** Flotillin expression is required for activation of mouse primary CD4 T cells. **a** Western blot of splenocyte lysate from WT and flotillin2 KO mice probed with antibodies against flotillin1 and flotillin2. **b–e** Representative flow cytometry histograms of CD4 T cells isolated from the spleen of WT or flotillin2 KO mice, activated on anti-CD3 and anti-CD28 surfaces for 20 and 36 h and stained with antibodies against CD69 (**b**) or CD25 (**d**). Mean fluorescent intensity (MFI) at the surface of 10,000 cells of WT and flotillin2 KO CD4 T cells stained with antibodies against CD69 (**c**) or CD25 (**e**). Data obtained from three mice for each condition. Error bars indicate mean (±SEM). *$p < 0.01$; **$p < 0.001$; ***$p < 0.0001$; ANOVA test

synapse. Hence, flotillins might further control TCR recycling by establishing a previously unobserved type of endosome.

In summary, we show that compartmentalization of TCR into a specialized sorting compartment demarked by flotillins controls recycling of TCR to the immunological synapse. Consistent with our findings, multiple evidence supports the idea that clathrin-independent endocytosis of surface proteins is connected to a specific recycling apparatus[55,56,58,60,61]. It would be interesting to determine if the endocytic network we uncovered contributes to the polarized recycling of cell surface proteins in other cell types during processes, such as cell migration[4], cell cytokinesis[5], and maintaining of basolateral polarity of epithelial cells[3]. Many questions remain as to which molecular mechanisms drive the clathrin-independent endocytosis of TCR, how the mobile endocytic network demarked by flotillin functions, and how TCR recycling actually supports signaling events and the nuclear translocation of transcription factors.

## Methods

**Mice**. Flot2$^{-/-}$ mice have been kindly provided by Tak W. Mak (The Campbell Family Institute for Breast Cancer Research, University of Toronto, Canada). C57BL/6J were maintained at the University of Konstanz. The organ collection was approved by the German Veterinary authority. Animal experiments were approved by the Review Board of Regierungspräsidium Freiburg in accordance with German Animal Protection Law (T-16/15TFA).

**Plasmids and CRISPR/Cas9**. Mammalian expression constructs encoding for Flotillin-1 or Flotillin-2 tagged with EGFP and mCherry were a gift from V. Nigli (Univerity of Bern). Clathrin-EGFP was kindly provided by T. Kirchausen

(Harvard Medical School). AP180-C-GFP was provided by H.T. McMahon (MRC Laboratory of Molecular Biology). Mammalian expression constructs encoding full-length wild-type human Lck was a gift from prof. T. Harder (University of Oxford). TCRζ-PS-CFP2 was provided by prof. K. Gaus (University of New South Wales). PA-mCherry expression backbone was obtained from Clontech.

TCRζ and Lck PA-mCherry were made by inserting a PCR product of TCR-PS-CFP2 into pPAmCherry-N1 using EcoRI + AgeI. Flotillin 1 and 2 PA-mCherry were made by inserting the PCR product from flotillin-EFGP into pPSCFP2-N1 using EcoRI + SacII and then by swapping PS-CFP2 for PA-mCherry using AgeI + NotI. PA-mCherry clathrin was made from EGFP-Clathrin by replacing EGFP with PA-mCherry using NheI + BglII.

For the knocking out of Flotillin-1 and -2, or LAT, Jurkat T cells were transfected with two guide RNAs that were specifically designed to target genomic DNA, together with Cas9 expression plasmid. Twenty-four hours post-transfection, transfected single cells were FACS sorted and seeded into 96-well plates. Cell clones were screened by using western blotting with flotillin (3253 and 3244S, 1:1000, Cell Signaling Technology) and Lat antibodies (9166, 1:1000, Cell Signaling Technology). Clones lacking flotillins or LAT eventually grown to an appropriate population in ~20 days.

**Cell culture and sample preparation**. Jurkat T cells (Clone E6.1), Jcam1, P116, Raji B-cells (all ATCC) all knock-out Jurkat cell lines and primary CD4 T cells were cultured in RPMI 1640 medium (Gibco) supplemented with 10% (vol/vol) FBS, and 2 mM L-Glutamine (all from Invitrogen). T cells were transfected with 1 μg DNA per 200,000 cells, 12–24 h prior to imaging using the Neon electroporation kit (Invitrogen).

Primary CD4+ T cells from spleen were isolated from wild-type C57BL/6J and flotillin2 knock-out mice using a CD4+ T cells Isolation Kit (130-049-201; Miltenyi Biotec), according to the manufacturer's instructions. Isolated CD4+ T cells were further activated on 5 μg/ml anti-CD3 and anti-CD28 surfaces (16-0033-82, 16-0281-82; eBioscience) in 96-well plates for the indicated time points.

Before imaging, cells were incubated for 10 min at 37 °C on 18 mm glass-coated surfaces (Marienfeld) that were prepared by over-night incubation at 4 °C of either

1 μM CD90 (14-0909-82, Jomar Bioscience) antibodies for resting conditions, or 1 μM CD3ε (16-0037; eBioscience) and CD28 (16-0289; eBioscience) antibodies for activating conditions. For live cell imaging, cells were imaged within 20 min after their deposition on the coverslips. For pitstop2 (Abcam) treatment, Jurkat T cells were seeded on 10 μg/ml anti-CD3ε-coated and anti-CD28 coated coverslips and 6 μM of the inhibitor was added 5 min after activation. T cells were allowed to activate for further 5–10 min before imaging. For dynasore and CK666 treatments, cells were activated on 10 μg/ml anti-CD3ε and anti-CD28 coverslips for 10 min and the inhibitor was added to cells at a concentration of 100 and 80 μM, respectively.

For photo-activation localization microscopy (PALM), cells were subsequently fixed in 3.7% EM-grade paraformaldehyde (C004, ProScitech) for 15 min at RT, washed, and resuspended in RPMI culture medium.

For transferrin internalization, transfected Jurkat T cells were washed once with 1xPBS, resuspended in serum-free OptiMEM medium (Sigma-Aldrich, USA), and placed on ice for 10 min to prevent endocytosis. Alexa488-conjugated transferrin (Jackson ImmunoResearch, USA) was added to T cells at 25 μg/ml (5 μl/ml) and incubated at 37 °C for 45 min. After incubation, cells were washed with 1xPBS, resuspended in OptiMEM medium, activated on 10 μg/ml anti-CD3ε and anti-CD28-coverslips for 10 min at 37 °C and imaged.

For immunostaining, cells were fixed with 3.7% EM-grade paraformaldehyde (C004, ProScitech) for 15 min at RT. After fixation, cells were permeabilized with 100 μg/ml lysolecithin, blocked in 5% BSA and probed with primary and secondary antibodies sequentially. We used DAPI (62248, 1:10,000, Thermo Fisher Scientific) for nuclei, phospho-TCRζ (Y142) (558486M, 1:20, BD Biosciences) with secondary antibody Donkey anti-Goat DyLight488 (ab96935; 1:200, Abcam), TSAd (bs-3858R; 1:100, Bioss antibodies), NFAT and phosphor-Zap70 (Y319) (D43B1 and 2701; 1:100 and 1:100, Cell Signaling Technologies) with secondary antibody Goat anti-Rabbit Alexa647 (111-606-047; 1:100, Jackson ImmunoResearch).

For the conjugation assay, $1 \times 10^6$/ml Raji B-cells were incubated with 2 μg/ml highly purified staphylococcal enterotoxins (SEE; Toxin Technology Inc.) for 1 h at 37 °C, washed, and pelleted (200 g, 5 min, 37 °C) with equal amount of Jurkat T cells (or derived knock-outs). Subsequently, the mixture of T-cells and B-cells is incubated for 10 min at 37 °C. Hereafter, the pellet was carefully resuspended and $\sim 0.5 \times 10^6$ cells were applied on a poly-L-lysine-coated coverslip. The cells were allowed to adhere on the coverslip prior to fixation with 3.7% (vol/vol) EM-grade paraformaldehyde (C004, ProScitech) for 15 min at RT. Poly-L-lysine coverslips were prepared by incubating clean 18 mm coverslips (Marienfield) for at least 1 h at 37 °C with 0.001% (vol/vol) poly-L-lysine (P4957; Sigma).

**Flow cytometry**. For the CD3-internalization assay, Jurkat T cells (WT or flotillin-1/-2 KO) were incubated with 1 μM biotinylated CD3ε (16-0037; eBioscience) and 1 μM non-biotinylated CD28 (16-0289; eBioscience) antibodies in solution at 4 °C to allow binding but not internalization. Upon removal of unbound antibodies by extensive washing of the cells with PBS, T cells were incubated for 5, 10, or 20 min at 37 °C. Antibody bound and surface-exposed CD3ε molecules were stained by Alexa647-streptavidin and the percentage of internalized CD3ε molecules is the mean fluorescent intensity (MFI) of timepoint 0 minus the MFI at the indicated timepoint.

To determine the recycling of CD3 molecules, T cells were incubated with 1 μM biotinylated CD3ε (16-0037; eBioscience) and 1 μM non-biotinylated CD28 (16-0289; eBioscience) antibodies in solution at 37 °C to allow binding and internalization. Subsequently, surface-exposed CD3 molecules were blocked by unlabelled streptavidin. Hereafter, re-surfacing of unblocked biotinylated CD3ε molecules was determined by staining with Alexa647-streptavidin after 5, 10, or 20 min incubation at 37 °C. The increased MFI relative to timepoint 0 correlates with the amount of recycled CD3 molecules.

In order to determine T cell receptor signaling, we activated Jurkat T cells (WT or Flotillin1/2 KO) for the indicated time with 1 μM biotinylated CD3ε (16-0037; eBioscience) and 1 μM non-biotinylated CD28 (16-0289; eBioscience) antibodies in solution. Subsequently, these activated cells were fixed with 3.7% (vol/vol) PFA (20 min RT) and permeabilized for 30 min on ice by slowly adding ice-cold methanol to a final concentration of 90% (vol/vol). Hereafter, the cells were stained with antibodies against phosphorylated TCRζ (558489; BD Phosflow), Zap70 (2704S; Cell Signaling Technologies), and ERK (9101S; Cell Signaling Technologies).

To quantify conjugate formation, Jurkat T cells and Raji B-cells were prepared as described above with minor modifications. T cells were stained with Fixable Violet Dead Stain kit (L-34963; Life Technologies) and B cells with carboxyfluorescein diacetate succinimidyl ester (CFDA; Invitrogen) prior to conjugation. Conjugates were defined as the double-positive population.

For the primary cell assay, cells were collected at the indicated time points and prepared for surface labelling. Cells were stained with BV-421 anti-mouse CD25 (Biolegend, Cat. 102033), APC anti-mouse CD3ε (BD Biosciences, Cat. 553066), and FITC anti-mouse CD69 (BD Biosciences, Cat. 557392). Samples were acquired using an LSR-II flow cytometer (Becton-Dickinson, San Jose, CA) and analyzed using FlowJo software 10 (Tree Star, Ashland, OR).

All flow-cytometry-based assays were performed on a BD FACSCanto II or a BD LSR II. Autofluorescence in a dump gate and the size of the population in FSC and SSC dot plot are used to distinguish live and dead cells. Doublets were eliminated by using a pulse geometry gate with FSC-H and FSC-A, except in the conjugation assay in which it confirmed formed conjugates.

**Microscopy**. Fixed and live-cell confocal microscopy were performed on a Zeiss LSM780 and/or LSM880 laser-scanning confocal microscope (Zeiss, Germany) that is equipped with an argon laser (405, 488 nm), a diode pump solid state laser (561, 647 nm), and a live-cell incubation chamber (Pecon). GFP constructs were exited using the 488 nm line of the argon laser source, while PA-mCherry tagged proteins were excited with the 561 nm laser line. Images were acquired with a 100 × 1.4NA DIC M27 Apo-Plan oil immersion objective (Zeis, Germany) and GaAsP-PMTs in simultaneous, bidirectional scanning mode, resulting in two-color frame recording almost every 10 s. For each channel the pinhole was set to 1 Airy Unit.

Live-cell TIRF and PALM images were acquired on a total internal reflection fluorescence microscope (ELYRA; Zeiss) with a 100 × oil-immersion objective with a numerical aperture of 1.46. For PS-CFP2, 8 μW of 405-nm laser radiation and imaging of green-converted PS-CFP2 with 18 mW of 488-nm light were used for photoconversion. For PALM, 20,000 images were acquired per sample with a cooled, electron-multiplying charge-coupled device camera (iXon DU-897D; Andor) with an exposure time of 18 ms. Recorded images were analyzed with Zeiss ZEN Black software. Drifting of the sample during acquisition was corrected relative to the position of surface-immobilized 200-nm colloidal gold beads (BBInternational) placed on each sample. Photoactivation was achieved by illuminating a subset of the cell outer membrane with a 7.2 μW 405 nm laser pulse with 12.24 μs per pixel dwell time.

**Optogenetic clustering**. Flotillin2-ECFP-CIB1 and flotillin2-mCerulean-Cry2Clust were designed based on Nguyen et al.[44] and Park et al.[45], respectively. Jurkat T cells were transfected with either flotillin2-ECFP-CIB1 or flotillin2-mCerulean-Cry2Clust 24 h prior to use. In a flat bottom 12-well plate (3513, Corning), transfected cells were exposed to 470 nm blue light from an LED tran-silluminator (LB-16, Maestro Gen) with a 10 s on/off pulse for 10 min at 37 °C prior to activation, following which 1 μl of anti-CD3ε and anti-CD28 were added to induce T cell activation. Blue light exposure was continued for a further 5 min, following which cells were transferred onto ice, pelleted and resuspended in cold 3.7% PFA and fixed on ice for 30 min. Untransfected WT and flotillin-KO Jurkat T cells were simultaneously activated for 5 min without blue light exposure, followed by fixation. Cells were then washed 3× in cold PBS and permeabilized with Cytoperm Plus (554715, BD Bioscience) for 10 min. Following fixation, cells were washed 1× with PBS and incubated with anti-phosphorylated TCR diluted in PBS + 5% BSA for 2 h at 4 °C. Cells were subsequently washed prior to flow cytometry.

Flow cytometry was performed on a BD FacsCantoX20. Intact sells were gated using FCS and SSC. Gating for transfected and anti-phosphorylated TCR647-stained cells was determined using untransfected, unstained cells, and adjusted using transfected only and activated, anti-phosphorylated TCR647-stained only cells. Cyan fluorescence was excited with 450 nm laser and detected using a blue/green 535 nm filter, while Alexa647 fluorescence was excited with 640 nm laser and detected with a red 670 nm filter.

**Imaging analysis**. Vesicle count & cross channel nearest neighbor distances: Images were analyzed with custom written Matlab vesicle tracking and cross-channel nearest-neighbor distance evaluation software. From the two-channel images, the EGFP channel served as a mask for the cell body, as well as marking the central endosomal region, if necessary for analysis, using user-set thresholds. Vesicles in both channels were separated by intensity-based thresholding from the background of the spatial band-pass-filtered image (pass band of 2–9 pixels) and localized. Counts in each channel and cross-channel nearest-neighbor distances for all vesicles were evaluated. Random cross-channel nearest-neighbor distances were calculated by shuffling the time information for the vesicle localizations before performing the nearest-neighbor calculations. A GUI application and all source code for this analysis is freely available from the author's GitHub repo at https://github.com/PRNicovich/PAVesT.git.

Nuclear translocation of transcription factors NFAT and TSAd were determined by using ImageJ/Fiji Plugin TANGO[62] and its included Nuclear Edge Detector for nuclear segmentation and determining integrated intensity density.

Vesicle tracking: we used DiaTrack 3.04 PRO[Vallotton et al Tritrack]. At first, we filtered the image with a Gaussian filter to obtain one kernel per vesicle. Secondly, we manually selected the region covering the inside of the cell. Hereafter, DiaTrack tracks the vesicles and tracks with appropriate speed and lifetime of at least five frames were selected. Displacement and speed was exported in.mat file into MATLAB R2014b (Mathworks) and relevant figures were generated from this data in Graphpad Prism 6.00 for Windows (GraphPad Software, La Jolla CA).

Vesicle membrane intensity measurements were performed using Fiji based on the channel containing the PA-mCherry-labeled proteins of interest. PA-region was determined in frame three (first frame after 405 nm pause-mediated fluorophore activation = 17 s after start of acquisition) and added to ROI manager. Average pixel intensity in PA region was measured at the indicated timepoints/frames. Intensities were normalized to the measured intensity of frame 3 (first frame after 405 nm pause-mediated fluorophore activation = 17 s after start of acquisition).

Plasma membrane mCherry-labeled protein intensity was measured with Fiji after background substraction in selected ROIs that mark the plasma membrane, a directly attached region underneath the surface ROI, and a region of the membrane opposite from the cell–cell interaction site (distal region). For comparison, immune

synapse and directly attached region intensities are normalized using the distal region its intensity.

Intensity compactness: The compactness of intensity was measured on the premise that the most compact a signal can be is if it were all concentrated within a single pixel. This measure ranges from 1, representing all the intensity residing within a single pixel, to 0, where the signal is equally distributed across all pixels.

$$\text{Intensity compactness} = \frac{\sum |x - \bar{x}|}{2\left(\sum x\right) - 2\bar{x}}$$ The single-molecule localization microscopy data was first analyzed with Zen 2012 Black edition software (Zeiss MicroImaging). After Gaussian and Laplace filtering, events were judged to have originated from single molecules when $I - M > 6S$, where $I$ is event intensity, $M$ is mean image intensity and $S$ the s.d. of image intensity. The center of each point-spread function was then calculated by fitting to a two-dimensional Gaussian distribution. After correction for sample drift with immobile 200 nm gold fiducial markers (BBInternational), the $x$–$y$ particle coordinates of each molecule were stored in a table. Two-dimensional molecular coordinates were cropped into nonoverlapping regions of 3 μm × 3 μm in area. It is known that individual fluorophores, both in PALM can 'blink', which results in multiple localizations of the same molecule. Therefore, we discard 'blinks' that are present within the same 100 nm circle region in more than five continuous frames or that reappear within 10 frames at this region.

We used DBSCAN analysis[41] to identify individual clusters within the 3 μm × 3 μm regions. The DBSCAN method detects clusters using a propagative method, which links points belonging to the same cluster based on two parameters; the minimum number of neighbors $\varepsilon$ ($\varepsilon = 3$) in the radius $r$ ($r = 20$ nm). The DBSCAN routine was implemented in MATLAB and subsequently coded in c++ and compiled in a MEX file (Matlab executable file) to improve the speed of processing as we were working with large data files. A GUI application and all source code for this analysis is freely available from the author's GitHub repo at: https://github.com/PRNicovich/ClusDoC.

**Statistical analysis**. All statistical analysis were performed using GraphPad software (Prism). Statistical significance between datasets was determined by performing two-tailed, unpaired non-parametric Mann–Whitney tests and ANOVA test. Graphs show mean values, and error bars represent the SEM. In statistical analysis, $p > 0.05$ is indicated as not significant (n.s.), whereas statistically significant values are indicated by asterisks as follows: $*p \leq 0.05$, $**p < 0.01$, $***p < 0.001$, and $****p < 0.0001$.

**Data availability**. Data are available on the figshare repository (https://doi.org/10.6084/m9.figshare.5974978.v1). MATLAB codes are available from the author's GitHub repo (https://github.com/PRNicovich/PAVesT.git and https://github.com/PRNicovich/ClusDoC).

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

## Acknowledgements

We would like to thank the staff of the BioMedical Imaging Facility of the University of New South Wales, Senthil Arumugam for helpful discussions, Tak Wah Mak for the flotillin2 knock-out mouse, Michael Basler for help with the mice, Massimo Pizzato for the AP180-C construct and the funding bodies: National Health, Medical Research Council (APP1102730), Australian Research Council (DE140101626), and Swiss National Science Foundation (31003A_172969).

## Author contributions

E.B.C. and F.K. performed experiments, analyzed the data, and contributed to write the manuscript; M.E., G.R., M.A., N.R., N.K.K., G.P.B.S., and H.V. performed and analyzed experiments; P.R.N. and M.C. established analysis; Q.D., Z.Y., and J.L. contributed to molecular biology. K.G. contributed to data interpretation and manuscript writing. J.R. designed the project and wrote the manuscript.

## Additional information

**Competing interests:** The authors declare no competing interests.

