## [Peer Review File · Nature Communications]

Reviewer #1 (Remarks to the Author):

This is an exciting paper that reexamines the recycling dynamics of the T cells receptor in the Jurkat cell lines and particularly focuses on the role of flotillins. The TCR is well described to undergo internalization and recycling in the steady state and during activation. Significant effort has been focused on this using flow cytometry or co-localization to identify important pathways. Surprisingly, there has been limited systematic dynamic analysis of TCR trafficking. The authors use a combination of photoactivation, pulse-chase surface labeling and super-resolution microscopy to examine the dynamics of TCR recycling and surface organization with and without intact flotillins and activation. The exciting finding is the flotillins are needed for TCR recycling in activated cells, augmented nanoscale clusters and signaling function. These molecules have been studied in the context of T cells signaling, but most of the prior studies focused on surrogate stimulation pathways and didn't generate a clear picture of how flotillins are functioning when the TCR is engaged. There are a few questions that need to be addressed before the results can be fully integrated.

1. There seems to be an issue with the "book-keeping" of the TCR in activated cells. They show that TCR levels on the surface are relatively stable in both WT and flotillin KO cells, that internalization is similar, but the recycling is much slower in KO than WT. If the last point is true, then how can the surface level of the TCR be similar over time in the WT and KO cells? In order to account for this there would need to be a large, slow recycling pool of TCR in the KO cells. Is this the case. Or how else can the results be reconciled.
2. The authors want to related the trafficking and signaling defects, but what is the evidence that the changes in nanoscale clustering and signaling are related to the recycling defect? It seems to me that since flotilling is associate with TCR in all compartments, at the plasma membrane, in the endocytic pathway and during recycling, the signaling defects could be unrelated to recycling defects, particularly as the recycling defect seems to have no impact on surface expression of the TCR, which is clearly the pool that needs to be involved in signaling. Can the authors do an experiments that shows that the recycling pool of receptors is the one that is responsible for the augmented nanoclusters and the signaling competence? Otherwise I think it would be acceptable to document the three defects and discuss the possible relationship, but also alternative interpretations where they are unrelated.
3. There is other evidence that independent vesicles trafficking systems are important for movement of TCR, LAT and Lck. This study, Soares et al, which is cited, focused on many trafficking regulators, but not flotillins. Can the authors discuss any possible links between the Rab, Mal and other proteins studied there and flotillins?

Minor points

1. The authors refer in many places to a "recycling cycle". This is an awkward phrase. Isn't it just "recycling".

2. The anti-CD90 as a resting substrate is surprising. Many of the early study on flotillins were focused on signaling in response to cross-linking GPI anchored proteins, which is known to activate T cells. So I would accept that anti-CD90 may produce activation without engaging the TCR, but I have a hard time accepting that it doesn't activate the Jurkat cells. This needs to at least be discussed. None of the papers cited to back this up actually use the anti-CD90 substrates. For example, were the Figure 7 d-g data acquired on anti-CD90 vs anti-CD3+anti-CD28 on substrates?

Reviewer #2 (Remarks to the Author):

This is a potentially interesting paper showing a role for flotillins in t-cell activation. However, there are some technical / interpretative issues that require significant further experiments.

1. The basic endocytosis assay used in Figure 1 is not sound as it does not discriminate between TCR in invaginations of the plasma membrane - which may extend some significant distance away from the coverslip in the imaging set-up used - and TCR in vesicles that are topologically fully resolved from the plasma membrane (pinched off).
2. Assertion that endocytosis is clathrin-independent is not sound as there is no positive control - pitstop should block Tf uptake. This experiment could be better performed with one of the well characterised dom -ve mutants that block clathrin function, such as the c-terminal domain of AP180.
3. The data on clustering/distribution of TCR in the flotillin KO cells are significantly more convincing than the endocytosis experiments

Reviewer #3 (Remarks to the Author):

In this report the authors describe a new clathrin-independent and flotillin-dependent mechanism which controls TCR recycling to the immune synapse (IS), impacting on TCR-dependent signaling and T cell activation.

Although TCR trafficking through the endocytic compartment has strongly emerged in recent years as a central player in IS assembly and function, the pathway that orchestrates sorting at the plasma membrane, endocytosis and recycling of the TCR are only beginning to be elucidated. In this respect, the data presented in this manuscript, where a photoactivation approach has been used to follow at

the single-cell level the dynamics of the TCR at the different steps of its journey from the plasma membrane to endosomes and back have significant elements of novelty. The results provide evidence that the TCR is associated with a new dynamic endocytic network marked by flotillins that regulates its nanoscale organization at the IS and promotes early and late signaling events essential to T cell activation.

Overall the experimental approach is sound and the data are solid. The work has however been entirely performed on Jurkat T cells. Although there are objective technical difficulties that preclude extending the majority of the experiments to primary T cells, the authors should attempt to validate at least the functional data (i.e. outcome of flotillin knockdown on T cell activation) on primary T cells. An endpoint such as expression of activation markers (e.g. CD69, CD25) in flotillin KO Jurkat cells should be moreover included.

Specific points

1. Figure 1. In this and subsequent figures the authors refer to "activated" cells to refer both to photoactivation of specific molecules (e.g. TCRzeta) and to cell activation by CD3/CD28 co-stimulation. This should be clarified in all figures.
2. Figure 1, panels g,h. Although the effects of dynasore and CK666 are clear, the authors should discuss the possible mechanisms involving dynamin and Arp2/3 that regulate the motility of the TCRzeta+ vesicles. Also, they propose that in the absence of functional dynamin or Arp2/3 the TCR remains associated with vesicles that fail to undergo fission and remain associated to the plasma membrane. This is not proven formally by the results, as the vesicles could alternatively pinch off but remain very close to the plasma membrane. The statement should be tuned down.
3. Figure 2. In panel b the max number of TCRzeta+ vesicles in untreated cells should be plotted as a dot distribution, similar to the Pitstop2-treated cells. In panel c the distribution of clathrin appears different in the cell expressing photoactivatable TCRzeta compared to the cell showing labelled transferrin. A similar consideration applies to flotillin in panel d. The authors should comment on this apparent discrepancy. I also suggest to use cells with similar size in both panel c and d (and add a size bar)
4. Figure 3, panel h. The constant increase of TCRzeta+ vesicles after repetitive photoactivation should be discussed in more detail.
5. Figure 4. The experiments should be extended to the transferrin receptor, which the authors show to be associated with clathrin (Fig.2) and should be therefore be unaffected by flotillin KO.
6. Figures 4 and 5. The authors state that flotillins1/2 KO inhibits TCR delivery to the IS (Fig.5) while not affecting TCR internalization and vesicle number, as assessed by measuring the number of TCRzeta+ vesicles 250 sec after photoactivation (Fig 4). I expect that at time points longer than 250 sec (for example 5 min, when the authors find some TCR recycling at the surface; Fig.4i), the number of TCRzeta+ vesicles would decrease due to their fusion with the plasma membrane while their

number does not change or even increases slightly in flotillin KO cells (Fig.4c; also here a dot distribution should be used for WT cells similar to the KO cells). The authors should show a longer time course of vesicle number tracking.

7. Figure 6, panel d. The 30 ± 4 nm population of TCRzeta+ clusters are clearly absent in activated flotillin1/2 KO cells. However it is not clear why the frequency of the 16 ± 2 nm cluster population is highly increased in flotillin1/2 KO cells compared to WT cells. The authors should comment on this point.

8. Figure 7, panels a and b. The authors describe the effects of flotillin 1/2 KO on activation-dependent T-cell signaling, using flow cytometry to measure the efficiency of conjugate formation and flow cytometry to quantify the extent of phosphorylation of key signaling molecules. I would suggest to measure some early signaling events (PTyr, P-ZAP-70, CD3 accumulation) in T cell-Raji cell conjugates to verify whether the conjugates that are formed are functional.

Minor points

1. Line 280: change TCRz with the corresponding symbol TCR ζ
2. In the legend to supplementary figure 2, the authors describe a panel "d" which is not shown in the figure.
3. In the legend to supplementary figure 4, the description of panel "b" is missing.
4. Line 313. Supplementary figure 8 is not included in the supplemental material. I guess that the authors are referring to supplementary figure 7.

Revision of " A mobile endocytic network connects clathrin-independent receptor endocytosis to recycling and promotes T cell activation" (NCOMMS-17-02117-T)

We thank the reviewers for their constructive questions and comments (included below in italic), which we have addressed point-by-point. Corresponding revisions are underlined in the revised manuscript. We have addressed all the points raised and hope that the reviewers agree that this revision has significantly strengthened the manuscript.

Reviewer #1

This is an exciting paper that reexamines the recycling dynamics of the T cells receptor in the Jurkat cell lines and particularly focuses on the role of flotillins. The TCR is well described to undergo internalization and recycling in the steady state and during activation. Significant effort has been focused on this using flow cytometry or co-localization to identify important pathways. Surprisingly, there has been limited systematic dynamic analysis of TCR trafficking. The authors use a combination of photoactivation, pulse-chase surface labeling and super-resolution microscopy to examine the dynamics of TCR recycling and surface organization with and without intact flotillins and activation. The exciting finding is the flotillins are needed for TCR recycling in activated cells, augmented nanoscale clusters and signaling function. These molecules have been studied in the context of T cells signaling, but most of the prior studies focused on surrogate stimulation pathways and didn't generate a clear picture of how flotillins are functioning when the TCR is engaged. There are a few questions that need to be addressed before the results can be fully integrated.

1. There seems to be an issue with the "book-keeping" of the TCR in activated cells. They show that TCR levels on the surface are relatively stable in both WT and flotillin KO cells, that internalization is similar, but the recycling is much slower in KO than WT. If the last point is true, then how can the surface level of the TCR be similar over time in the WT and KO cells? In order to account for this there would need to be a large, slow recycling pool of TCR in the KO cells. Is this the case. Or how else can the results be reconciled.

We thank the reviewer for this insightful hypothesis, which we tested and found to be correct. Impaired TCR recycling in flotillin1/2 KO cells did result in an increased number of TCR ζ -positive endocytic/recycling vesicles 10 min after photoactivation (*i.e.* internalization; Fig. 4j and k, new panels). This observation suggests that TCR ζ indeed accumulated in intracellular compartments in the absence of flotillin. We have modified the manuscript to include this observation (lines 273-279).

2. The authors want to related the trafficking and signaling defects, but what is the evidence that the changes in nanoscale clustering and signaling are related to the recycling defect? It seems to me that since flotilling is associate with TCR in all compartments, at the plasma membrane, in the endocytic pathway and during recycling, the signaling defects could be unrelated to recycling defects, particularly as the recycling defect seems to have no impact on surface expression of the TCR, which is clearly the pool that needs to be involved in signaling. Can the authors do an experiments that shows that the recycling pool of receptors is the one that is responsible for the augmented nanoclusters and the signaling competence?

Otherwise I think it would be acceptable to document the three defects and discuss the possible relationship, but also alternative interpretations where they are unrelated.

The submitted manuscript does indeed not show evidence that the difference in TCR clustering is directly related to the recycling defect. The fact that a defect in recycling might affect the spatial organization of TCR at the plasma membrane was only our favored hypothesis. The heading of the related result paragraph could have been misleading and we apologize for the possible confusion. We have amended the title and the paragraph itself to avoid favoring one potential mechanism over the other (lines 316-318 and 333-335)

We nevertheless tried to perform an experiment, as suggested by the reviewer, to show a direct connection between recycling and TCR clustering. We used a combination of antibody feeding assay and single molecule localization microscopy (dSTORM) to visualize clusters of TCR ζ resulting from recycling. Unfortunately, dSTORM requires a strong signal to be properly quantitative and the signal generated by recycled TCR was too weak, which did not allow quantification.

In the original submission, we also discussed the possibility that flotillins contribute to delineate domains at the surface of intracellular compartments and how these interactions could support recycling. We have now modified this part of the discussion to include the possibility of flotillins directly organizing TCR within the plasma membrane as well. We do not think that a potential flotillin-mediated organization of TCR contributes to its endocytosis because our data conclusively show that flotillins did not support TCR internalization (Fig. 4 a-f). The original and amended text can be found in the discussion at lines 436-445

3. There is other evidence that independent vesicles trafficking systems are important for movement of TCR, LAT and Lck. This study, Soares et al, which is cited, focused on many trafficking regulators, but not flotillins. Can the authors discuss any possible links between the Rab, Mal and other proteins studied there and flotillins?

The study of Soares *et al.* indeed has similar conclusions to the submitted manuscript regarding the importance of intracellular trafficking for T cell activation. However, there is to date no connection between flotillins and the Rab proteins identified in this study as contributing to TCR transport. To our knowledge, neither Rab4b, Rab3d nor Rab8b have been shown to have any link with flotillins. Only one Rab protein, Rab11a, has been clearly identified as being connected to flotillins (Bodrikov *et al.*, 2017; Hülbusch *et al.*, 2015). This connection was demonstrated in the context of targeted recycling, which suggests that it is likely to happen in T cells as well, albeit Soares *et al* do not mention Rab11a. We are currently investigating this possibility in a follow-up study. In the same work, we also plan to investigate a potential link between Rab4b, Rab3d or Rab8b and flotillins in T cells, but this falls beyond the scope of the present manuscript.

As for MAL, it has been shown to regulate the trafficking of Lck towards the immunological synapse (Antón *et al.*, 2008), but not that of TCR. In our hands, the intracellular trafficking of Lck was very different from that of TCR (Fig.1), similarly to what has been observed by Soares *et al.* Thus, the MAL pathway is unlikely to be linked with flotillin-mediated recycling, but this remains to be investigated in future studies.

Minor points

1. The authors refer in many places to a “recycling cycle”. This is an awkward phrase. Isn't it just “recycling”.

We have replaced “recycling cycle” by “recycling” wherever it appeared in the manuscript.

2. The anti-CD90 as a resting substrate is surprising. Many of the early study on flotillins were focused on signaling in response to cross-linking GPI anchored proteins, which is known to activate T cells. So I would accept that anti-CD90 may produce activation without engaging the TCR, but I have a hard time accepting that it doesn't activate the Jurkat cells. This needs to at least be discussed. None of the papers cited to back this up actually use the anti-CD90 substrates. For example, were the Figure 7 d-g data acquired on anti-CD90 vs anti-CD3+anti-CD28 on substrates?

Experiments were conducted in a previous study in order to demonstrate that Jurkat T cells are not activated upon binding onto CD90-coated surfaces. Supplementary Fig. 7 of (Ma *et al.*, 2017) shows that CD90 does not trigger calcium flux or CD69 expression. This study was not published at the time of the initial submission of the manuscript and we could therefore not cite it. We have now amended the manuscript and cite it (reference 38, line 321).

References:

Antón, O., Batista, A., Millán, J., Andrés-Delgado, L., Puertollano, R., Correas, I., and Alonso, M.A. (2008). An essential role for the MAL protein in targeting Lck to the plasma membrane of human T lymphocytes. *J. Exp. Med.* 205, 3201–3213.

Bodrikov, V., Pauschert, A., Kochlamazashvili, G., and Stuermer, C.A.O.O. (2017). Reggie-1 and reggie-2 (flotillins) participate in Rab11a-dependent cargo trafficking, spine synapse formation and LTP-related AMPA receptor (GluA1) surface exposure in mouse hippocampal neurons. *Exp. Neurol.* 289, 31–45.

Hülsbusch, N., Solis, G.P., Katanaev, V.L., and Stuermer, C.A.O. (2015). Reggie-1/Flotillin-2 regulates integrin trafficking and focal adhesion turnover via Rab11a. *Eur. J. Cell Biol.* 94, 531–545.

Ma, Y., Pandzic, E., Nicovich, P.R., Yamamoto, Y., Kwiatek, J., Paeon, S. V., Benda, A., Rosy, J., and Gaus, K. (2017). An intermolecular FRET sensor detects the dynamics of T cell receptor clustering. *Nat. Commun.* 8, 15100.

Reviewer #2

This is a potentially interesting paper showing a role for flotillins in t-cell activation. However, there are some technical / interpretative issues that require significant further experiments.

1. The basic endocytosis assay used in Figure 1 is not sound as it does not discriminate between TCR in invaginations of the plasma membrane - which may extend some significant distance away from the coverslip in the imaging set-up used - and TCR in vesicles that are topologically fully resolved from the plasma membrane (pinched off).

The photoactivation approach would indeed not be able to discriminate between invaginations of the plasma membrane and endocytic vesicles in a system where these vesicles would remain static after pinching off the plasma membrane. However, the strength of our approach is the ability to track the dynamics and intracellular movement of internalized proteins. In the cells and conditions described in the submitted manuscript, endocytosis of all PA-mCherry-tagged fusion proteins (TCR ζ , flotillins, clathrin) was systematically concurrent with rapid and sustained movements of the internalized vesicles. This observation would not be compatible with endocytic pits or invaginations that would not have pinched off of the plasma membrane, which would be observed as static over the timescales of our experiments. In fact, the vesicles were static only when the pinching off was prevented by dynamin or Arp2/3 inhibition (Fig. 1 e-h), providing a way to discriminate between complete endocytosis followed by intracellular trafficking and accumulation in endocytic pits.

2. Assertion that endocytosis is clathrin-independent is not sound as there is no positive control - pitstop should block tf uptake. This experiment could be better performed with one of the well characterised dom -ve mutants that clock clathrin function, such as the c-terminal domain of AP180.

We fully agree with the reviewer on the importance of performing proper positive controls regarding the effect of Pitstop2. As suggested, we have assessed the effect of Pitstop2 on transferrin uptake using Alexa488-labelled transferrin. We could not observe internalization of transferrin-Alexa488 in the presence of 6 or 12 μ M of Pitstop2, even after 30 min at 37C. Representative images of these experiments are now shown in Supplementary Fig 2d (and mentioned in the main text at line 124). This result is also in accordance with the data presented in the original manuscript in Supplementary Fig 2c, showing that cells treated with Pitstop2 form significantly less clathrin-coated vesicles than their untreated counterparts.

We have further followed the reviewer's suggestion to use the c-terminal domain of AP180 as an inhibitor of clathrin-mediated endocytosis. In accordance with the data obtained using Pitstop2, internalization of TCR ζ -PA-mCherry was not impaired in cells overexpressing c-term-AP180. These data have been added to Fig.2 (panels a and b) and are mentioned in the text at lines 128-132.

In summary:

- 1) Internalization of TCR ζ was not affected by Pitstop2, which, in the same conditions, prevented the formation of clathrin-coated vesicles and abolished transferrin internalization.
- 2) Internalization of TCR ζ was not affected by the expression of to the dominant negative c-terminal domain of AP180.
- 3) In Jurkat T cells, inhibition of Arp2/3 activity with CK666 prevented pinching off of TCR ζ - and flotillin-positive vesicles, but had no effect on clathrin-coated vesicles.

Taken together, these results allow us to confidently state that TCR ζ is not internalized through clathrin-coated vesicles in activated Jurkat T cells.

Reviewer #3

In this report the authors describe a new clathrin-independent and flotillin-dependent mechanism which controls TCR recycling to the immune synapse (IS), impacting on TCR-dependent signaling and T cell activation.

Although TCR trafficking through the endocytic compartment has strongly emerged in recent years as a central player in IS assembly and function, the pathway that orchestrates sorting at the plasma membrane, endocytosis and recycling of the TCR are only beginning to be elucidated. In this respect, the data presented in this manuscript, where a photoactivation approach has been used to follow at the single-cell level the dynamics of the TCR at the different steps of its journey from the plasma membrane to endosomes and back have significant elements of novelty. The results provide evidence that the TCR is associated with a new dynamic endocytic network marked by flotillins that regulates its nanoscale organization at the IS and promotes early and late signaling events essential to T cell activation.

Overall the experimental approach is sound and the data are solid. The work has however been entirely performed on Jurkat T cells. Although there are objective technical difficulties that preclude extending the majority of the experiments to primary T cells, the authors should attempt to validate at least the functional data (i.e. outcome of flotillin knockdown on T cell activation) on primary T cells. An endpoint such as expression of activation markers (e.g. CD69, CD25) in flotillin KO Jurkat cells should be moreover included.

We have obtained a flotillin2 knock-out mouse from the group of T Mak (Berger et al., 2012) to perform the suggested experiments on primary T cells. The lack of flotillin2 expression in splenocytes isolated from these mice resulted in a complete downregulation of flotillin1, making these mice a *de facto* flotillin double knock-out (Fig. 8a).

CD4 T cells isolated from the spleen of these mice and stimulated on surfaces coated with anti-CD3 and anti-CD28 had significantly less CD69 and CD25 surface expression after 20h and 36h of stimulation respectively (Fig. 8c and b, and lines 373-380 in the main text)

Specific points

1. *Figure 1. In this and subsequent figures the authors refer to "activated" cells to refer both to photoactivation of specific molecules (e.g. TCRzeta) and to cell activation by CD3/CD28 co-stimulation. This should be clarified in all figures.*

We apologize for the confusing labeling in figures 1 and 2. We now use “activated” for CD3/CD28 co-stimulation and “PAct.” for photoactivation.

2. *Figure 1, panels g,h. Although the effects of dynasore and CK666 are clear, the authors should discuss the possible mechanisms involving dynamin and Arp2/3 that regulate the motility of the TCRzeta+ vesicles. Also, they propose that in the absence of functional dynamin or Arp2/3 the TCR remains associated with vesicles that fail to undergo fission and remain associated to the plasma membrane. This is not proven formally by the results, as the vesicles could alternatively pinch off but remain very close to the plasma membrane. The statement should be tuned down.*

The reviewer is correct, we did not formally show that TCR vesicles are still bound to the plasma membrane in dynasore and CK666 treated T cells (by, for instance, using electron microscopy). We have moderated the text as suggested (lines 103 and 112-118).

We nevertheless would like to clarify why we think that the immobility of TCR endocytic vesicles is related to an impaired pinching off of the plasma membrane rather than to ineffective transport: a) the static vesicles got brighter with time (Fig. 1g and h). This could be explained by the fact they keep accumulating TCR ζ -PAmCherry molecules coming from the photoactivated region of interest at the plasma membrane, which would be possible only if they were still connected to the plasma membrane. b) imaging of tubulin and TCR ζ vesicles as shown in Supplementary Fig. 4a suggests that these vesicles are transported along microtubules and do not rely on actin branching for their movement. c) previous studies report that flotillin endosomes rely on microtubules for their mobility rather than actin (Cornfine *et al.*, 2011; Solis *et al.*, 2013).

3. *Figure 2. In panel b the max number of TCRzeta+ vesicles in untreated cells should be plotted as a dot distribution, similar to the Pitstop2-treated cells. In panel c the distribution of clathrin appears different in the cell expressing photoactivatable TCRzeta compared to the cell showing labelled transferrin. A similar consideration applies to flotillin in panel d. The authors should comment on this apparent discrepancy. I also suggest to use cells with similar size in both panel c and d (and add a size bar)*

We have modified Figure 2b, c, and d as suggested. We apologize for the mistaken attribution of some of the colors in Figure 2c and d.

To clarify however, the distribution of clathrin is indeed different between the two images in figure 2c. This is mainly because these are two different “pools” of clathrin: 1) on cells expressing TCR ζ -PAmCherry (on the left), all the exogenous clathrin in the cells is visible, as it is fused to EGFP; 2) in cells incubated with labelled transferrin (on the right), only the

photoactivated subpopulation of clathrin PA-mCherry is visible. It is the same for flotillin in panel d. We did not observe any effect of TCR ζ -PA-mCherry expression on the distribution of clathrin or flotillin positive vesicles. Nonetheless, we have followed the reviewer's suggestion and tried to show cells with more consistent appearance and size.

4. Figure 3, panel h. The constant increase of TCRzeta+ vesicles after repetitive photoactivation should be discussed in more detail.

The constant increase of TCR ζ positive vesicles after repetitive photoactivation is now discussed in more detail in the main text (lines 200-205). This result suggests that TCR ζ -PA-mCherry molecules diffused fast enough to at least partially replete the photoactivated region of interest within the 8 sec between the photoactivation pulses. The steady number of vesicles for 450 sec following one activation and the increasing number of vesicles resulting from repetitive photoactivations indicate that, following internalization, TCR ζ is only returned slowly to the plasma membrane and remains for a significant time in endocytic compartments.

5. Figure 4. The experiments should be extended to the transferrin receptor, which the authors show to be associated with clathrin (Fig.2) and should be therefore be unaffected by flotillin KO.

Fusing the transferrin receptor with PA-mCherry unfortunately resulted in an almost total inhibition of the PA-mCherry photoactivation. Instead, we used transferrin labelled with Alexa488 to visualize clathrin-dependent endocytosis as in Fig. 2. We could not detect any difference of transferrin endocytosis between WT and flotillin1/2 KO cells in the formation of endocytic vesicles containing transferrin-Alexa488 (Supplementary Fig. 5b-d and lines 236-237 in the main text).

6. Figures 4 and 5. The authors state that flotillins1/2 KO inhibits TCR delivery to the IS (Fig.5) while not affecting TCR internalization and vesicle number, as assessed by measuring the number of TCRzeta+ vesicles 250 sec after photoactivation (Fig 4). I expect that at time points longer than 250 sec (for example 5 min, when the authors find some TCR recycling at the surface; Fig.4i), the number of TCRzeta+ vesicles would decrease due to their fusion with the plasma membrane while their number does not change or even increases slightly in flotillin KO cells (Fig.4c; also here a dot distribution should be used for WT cells similar to the KO cells). The authors should show a longer time course of vesicle number tracking.

We have performed a longer time-course as suggested by the reviewer and observed significantly more TCR ζ -PA-mCherry positive vesicles 10 min after internalization in the flotillin1/2 KO cells than in WT cells (Fig. 4 j,k). This evidence suggests that internalized TCR ζ fails to recycle properly and accumulates in intracellular compartments in the absence of flotillins. We have modified the manuscript to include this observation (lines 273-279).

The data also suggest a decrease in the number of TCR ζ -PA-mCherry positive vesicles between 1 and 10 min for the WT and an increase for the flotillin1/2 KO cells, although these changes were not significant (see Fig. R1 just below). A decrease in TCR containing vesicles in WT cells would suggest that these vesicles start fusing back to the plasma membrane.

Figure R1: Comparison of the number of endocytic vesicles detected 1 and 10 min after internalization in WT and flotillin1/2 KO cells.

7. Figure 6, panel d. The 30 ± 4 nm population of TCR ζ + clusters are clearly absent in activated flotillin1/2 KO cells. However it is not clear why the frequency of the 16 ± 2 nm cluster population is highly increased in flotillin1/2 KO cells compared to WT cells. The authors should comment on this point.

The 16 ± 2 nm population is at the limit of what our assay can detect and we do not feel confident to draw definitive conclusions from the very small changes that can – or cannot – be observed in Fig. 6d. Our interpretation is that there is no significant change in this population, which would suggest that the larger 30 ± 4 nm clusters do not result from the concatenation of the smaller 16 ± 2 nm. We have added a sentence in the manuscript to comment on this point (lines 326-330).

8. Figure 7, panels a and b. The authors describe the effects of flotillin 1/2 KO on activation-dependent T-cell signaling, using flow cytometry to measure the efficiency of conjugate formation and flow cytometry to quantify the extent of phosphorylation of key signaling molecules. I would suggest to measure some early signaling events (pTyr, P-ZAP-70, CD3 accumulation) in T cell-Raji cell conjugates to verify whether the conjugates that are formed are functional.

As suggested by the reviewer, we have stained T cell-Raji B cell conjugates for pTCR ζ and pZap70 to verify if the conjugates formed between flotillin1/2 KO cells and Raji B cells were functional. These immunofluorescence results showed a similar trend as the data obtained by flow cytometry following anti-CD3 and anti-CD28 stimulation. We consistently observed a moderate decrease of pTCR ζ and pZap70 in the flotillin1/2 KO cells when compared to the WT cells, indicating that the conjugates formed by the flotillin1/2 KO displayed impaired signaling but were at least partially functional. These data can be found in Supplementary Fig. 8 and are mentioned in lines 363-366 in the main text.

Minor points

1. *Line 280: change TCRz with the corresponding symbol TCR ζ*
2. *In the legend to supplementary figure 2, the authors describe a panel “d” which is not shown in the figure.*
3. *In the legend to supplementary figure 4, the description of panel “b” is missing.*
4. *Line 313. Supplementary figure 8 is not included in the supplemental material. I guess that the authors are referring to supplementary figure 7.*

We thank the reviewer for bringing these mistakes to our attention and have corrected the manuscript accordingly.

References:

- Berger, T., Ueda, T., Arpaia, E., Chio, I.I.C., Shirdel, E.A., Jurisica, I., Hamada, K., You-Ten, A., Haight, J., Wakeham, A., et al. (2012). Flotillin-2 deficiency leads to reduced lung metastases in a mouse breast cancer model. *Oncogene* 32, 4989–4994.
- Cornfine, S., Himmel, M., Kopp, P., El Azzouzi, K., Wiesner, C., Krüger, M., Rudel, T., and Linder, S. (2011). The kinesin KIF9 and reggie/flotillin proteins regulate matrix degradation by macrophage podosomes. *Mol. Biol. Cell* 22, 202–215.
- Solis, G.P., Hülsbusch, N., Radon, Y., Katanaev, V.L., Plattner, H., and Stuermer, C.A.O. (2013). Reggies/flotillins interact with Rab11a and SNX4 at the tubulovesicular recycling compartment and function in transferrin receptor and E-cadherin trafficking. *Mol. Biol. Cell* 24, 2689–2702.

Reviewer #1 (Remarks to the Author):

The authors have addressed by concerns and made appropriate modification to the text. I have no further concerns.

Reviewer #2 (Remarks to the Author):

The authors have addressed my previous criticisms. I point out that they have not proved that it is the recycling of TCR that is responsible for the flotillin-dependent T-cell activation phenotypes, but think that nevertheless the ms will be of interest to people working in the relevant fields.

Reviewer #3 (Remarks to the Author):

The results presented in this report provide evidence that the TCR is associated with a new dynamic endocytic network marked by flotillins that regulates its nanoscale organization at the immune synapse and promotes early and late signaling events essential to T cell activation. The data have significant elements of novelty. The experimental approach is sound and the data are solid. The authors have satisfactory addressed all the issues raised in my previous review, adding substantial additional experimental evidence to support their findings.

2nd Revision of "A mobile endocytic network connects clathrin-independent receptor endocytosis to recycling and promotes T cell activation" (NCOMMS-17-02117B)

We thank the reviewers for their enthusiastic comments (included below in italic). We have addressed the additional point raised by reviewer 2 (underlined in the revised main text) and trust that this revision has rectified any possible confusion regarding the novelties presented in this work.

Reviewer #1

The authors have addressed by concerns and made appropriate modification to the text. I have no further concerns.

We thank the reviewer for the positive comment and for acknowledging the modification made to the text.

Reviewer #2

The authors have addressed my previous criticisms. I point out that they have not proved that it is the recycling of TCR that is responsible for the flotillin-dependent T-cell activation phenotypes, but think that nevertheless the ms will be of interest to people working in the relevant fields.

We thank the reviewer for acknowledging that we have addressed their original concerns and that the manuscript is of interest.

We apologise for the misunderstanding regarding the new point raised by the reviewer. It was not our intention to state that we have demonstrated a direct and irrefutable link between the recycling mediated by flotillins and T cell activation. It was only our favoured hypothesis, considering **a)** published evidence showing that endocytic recycling of TCR is required for T cell activation¹⁻¹⁰ and the data presented in this manuscript showing that flotillins are required for both **b)** TCR recycling and **c)** T cell activation.

We have nevertheless tried to include further experiments supporting our hypothesis. Solely inhibiting the contribution of flotillins to recycling while preserving their other potential contributions to T cell activation proved to be challenging. We have designed an experimental protocol, adapting an optogenetic tool recently published^{11,12} to artificially aggregate and disable flotillin-positive endosomes. The percentage of Jurkat T cells positive for phosphorylated TCR in response to activation was reduced upon aggregation of flotillin positive endosomes (Supplementary figure 9 and lines 372-377), suggesting a link between flotillin-mediated recycling and T cell activation.

We are aware that this new data is not evidence that it is only the absence flotillin-mediated recycling that results in impaired T cell activation observed in flotillin knockout T cells. We have therefore amended the original text of the manuscript to avoid possible any misunderstanding regarding what we have experimentally demonstrated (lines 15, 53 and 420).

References:

1. Onnis, A., Finetti, F. & Baldari, C. T. Vesicular Trafficking to the Immune Synapse: How to Assemble Receptor-Tailored Pathways from a Basic Building Set. *Front. Immunol.* **7**, 50 (2016).
2. Soares, H., Lasserre, R. & Alcover, A. Orchestrating cytoskeleton and intracellular vesicle traffic to build functional immunological synapses. *Immunol. Rev.* **256**, 118–32 (2013).
3. Monjas, A., Alcover, A. & Alarcón, B. Engaged and Bystander T Cell Receptors Are Down-modulated by Different Endocytotic Pathways. *J. Biol. Chem.* **279**, 55376–84 (2004).
4. Barr, V. A. *et al.* T-cell antigen receptor-induced signaling complexes: internalization via a cholesterol-dependent endocytic pathway. *Traffic* **7**, 1143–62 (2006).
5. Liu, H., Rhodes, M., Wiest, D. L. & Vignali, D. A. . On the Dynamics of TCR:CD3 Complex Cell Surface Expression and Downmodulation. *Immunity* **13**, 665–675 (2000).
6. Das, V. *et al.* Activation-induced polarized recycling targets T cell antigen receptors to the immunological synapse; involvement of SNARE complexes. *Immunity* **20**, 577–88 (2004).
7. Soares, H. *et al.* Regulated vesicle fusion generates signaling nanoterritories that control T cell activation at the immunological synapse. *J. Exp. Med.* **210**, 2415–33 (2013).
8. Blas-Rus, N. *et al.* Aurora A drives early signalling and vesicle dynamics during T-cell activation. *Nat. Commun.* **7**, 11389 (2016).
9. Finetti, F. *et al.* Intraflagellar transport is required for polarized recycling of the TCR/CD3 complex to the immune synapse. *Nat Cell Biol* **11**, 1332–1339 (2009).
10. Osborne, D. G., Piotrowski, J. T., Dick, C. J., Zhang, J.-S. & Billadeau, D. D. SNX17 Affects T Cell Activation by Regulating TCR and Integrin Recycling. *J. Immunol.* **194**, (2015).
11. Nguyen, M. K. *et al.* Optogenetic oligomerization of Rab GTPases regulates intracellular membrane trafficking. *Nat. Chem. Biol.* **12**, 431–436 (2016).
12. Park, H. *et al.* Optogenetic protein clustering through fluorescent protein tagging and extension of CRY2. *Nat. Commun.* **8**, 30 (2017).

Reviewer #3

The results presented in this report provide evidence that the TCR is associated with a new dynamic endocytic network marked by flotillins that regulates its nanoscale organization at the immune synapse and promotes early and late signaling events essential to T cell activation. The data have significant elements of novelty. The experimental approach is sound and the data are solid. The authors have satisfactorily addressed all the issues raised in my previous review, adding substantial additional experimental evidence to support their findings.

We thank the reviewer for their comment and for highlighting both the novelty and the solidity of the data presented in the manuscript, as well as the substantial additional experimental evidence provided in the first revision.

Reviewer #2 (Remarks to the Author):

The paper is now, in my opinion, suitable for publication

A mobile endocytic network connects clathrin-independent receptor endocytosis to recycling and promotes T cell activation

Point-by-point response to referees' comments

First Revision:

Reviewer #1

This is an exciting paper that reexamines the recycling dynamics of the T cells receptor in the Jurkat cell lines and particularly focuses on the role of flotillins. The TCR is well described to undergo internalization and recycling in the steady state and during activation. Significant effort has been focused on this using flow cytometry or co-localization to identify important pathways. Surprisingly, there has been limited systematic dynamic analysis of TCR trafficking. The authors use a combination of photoactivation, pulse-chase surface labeling and super-resolution microscopy to examine the dynamics of TCR recycling and surface organization with and without intact flotillins and activation. The exciting finding is the flotillins are needed for TCR recycling in activated cells, augmented nanoscale clusters and signaling function. These molecules have been studied in the context of T cells signaling, but most of the prior studies focused on surrogate stimulation pathways and didn't generate a clear picture of how flotillins are functioning when the TCR is engaged. There are a few questions that need to be addressed before the results can be fully integrated.

1. There seems to be an issue with the "book-keeping" of the TCR in activated cells. They show that TCR levels on the surface are relatively stable in both WT and flotillin KO cells, that internalization is similar, but the recycling is much slower in KO than WT. If the last point is true, then how can the surface level of the TCR be similar over time in the WT and KO cells? In order to account for this there would need to be a large, slow recycling pool of TCR in the KO cells. Is this the case. Or how else can the results be reconciled.

We thank the reviewer for this insightful hypothesis, which we tested and found to be correct. Impaired TCR recycling in flotillin1/2 KO cells did result in an increased number of TCR ζ -positive endocytic/recycling vesicles 10 min after photoactivation (*i.e.* internalization; Fig. 4j and k, new panels). This observation suggests that TCR ζ indeed accumulated in intracellular compartments in the absence of flotillin. We have modified the manuscript to include this observation (lines 273-279).

2. The authors want to related the trafficking and signaling defects, but what is the evidence that the changes in nanoscale clustering and signaling are related to the recycling defect? It seems to me that since flotilling is associate with TCR in all compartments, at the plasma membrane, in the endocytic pathway and during recycling, the signaling defects could be unrelated to recycling defects, particularly as the recycling defect seems to have no impact on surface expression of the TCR, which is clearly the pool that needs to be involved in signaling. Can the authors do an experiments that shows that the recycling pool of receptors is the one that is responsible for the augmented nanoclusters and the signaling competence? Otherwise I think it would be acceptable to document the three defects and discuss the possible relationship, but also alternative interpretations where they are unrelated.

The submitted manuscript does indeed not show evidence that the difference in TCR clustering is directly related to the recycling defect. The fact that a defect in recycling might affect the spatial organization of TCR at the plasma membrane was only our favored hypothesis. The heading of the related result paragraph could have been misleading and we apologize for the possible confusion. We have amended the title and the paragraph itself to avoid favoring one potential mechanism over the other (lines 316-318 and 333-335)

We nevertheless tried to perform an experiment, as suggested by the reviewer, to show a direct connection between recycling and TCR clustering. We used a combination of antibody feeding assay and single molecule localization microscopy (dSTORM) to visualize clusters of TCR ζ resulting from recycling. Unfortunately, dSTORM requires a strong signal to be properly quantitative and the signal generated by recycled TCR was too weak, which did not allow quantification.

In the original submission, we also discussed the possibility that flotillins contribute to delineate domains at the surface of intracellular compartments and how these interactions could support recycling. We have now modified this part of the discussion to include the possibility of flotillins directly organizing TCR within the plasma membrane as well. We do not think that a potential flotillin-mediated organization of TCR contributes to its endocytosis because our data conclusively show that flotillins did not support TCR internalization (Fig. 4 a-f). The original and amended text can be found in the discussion at lines 436-445

3. There is other evidence that independent vesicles trafficking systems are important for movement of TCR, LAT and Lck. This study, Soares et al, which is cited, focused on many trafficking regulators, but not flotillins. Can the authors discuss any possible links between the Rab, Mal and other proteins studied there and flotillins?

The study of Soares *et al.* indeed has similar conclusions to the submitted manuscript regarding the importance of intracellular trafficking for T cell activation. However, there is to date no connection between flotillins and the Rab proteins identified in this study as contributing to TCR transport. To our knowledge, neither Rab4b, Rab3d nor Rab8b have been shown to have any link with flotillins. Only one Rab protein, Rab11a, has been clearly identified as being connected to flotillins (Bodrikov *et al.*, 2017; Hülbusch *et al.*, 2015). This connection was demonstrated in the context of targeted recycling, which suggests that it is likely to happen in T cells as well, albeit Soares *et al* do not mention Rab11a. We are currently investigating this possibility in a follow-up study. In the same work, we also plan to investigate a potential link between Rab4b, Rab3d or Rab8b and flotillins in T cells, but this falls beyond the scope of the present manuscript.

As for MAL, it has been shown to regulate the trafficking of Lck towards the immunological synapse (Antón *et al.*, 2008), but not that of TCR. In our hands, the intracellular trafficking of Lck was very different from that of TCR (Fig.1), similarly to what has been observed by Soares *et al.* Thus, the MAL pathway is unlikely to be linked with flotillin-mediated recycling, but this remains to be investigated in future studies.

Minor points

1. The authors refer in many places to a “recycling cycle”. This is an awkward phrase. Isn’t it just “recycling”.

We have replaced “recycling cycle” by “recycling” wherever it appeared in the manuscript.

2. The anti-CD90 as a resting substrate is surprising. Many of the early study on flotillins were focused on signaling in response to cross-linking GPI anchored proteins, which is known to activate T cells. So I would accept that anti-CD90 may produce activation without engaging the TCR, but I have a hard time accepting that it doesn’t activate the Jurkat cells. This needs to at least be discussed. None of the papers cited to back this up actually use the anti-CD90 substrates. For example, were the Figure 7 d-g data acquired on anti-CD90 vs anti-CD3+anti-CD28 on substrates?

Experiments were conducted in a previous study in order to demonstrate that Jurkat T cells are not activated upon binding onto CD90-coated surfaces. Supplementary Fig. 7 of (Ma *et al.*, 2017) shows that CD90 does not trigger calcium flux or CD69 expression. This study was not published at the time of the initial submission of the manuscript and we could therefore not cite it. We have now amended the manuscript and cite it (reference 38, line 321).

References:

Antón, O., Batista, A., Millán, J., Andrés-Delgado, L., Puertollano, R., Correas, I., and Alonso, M.A. (2008). An essential role for the MAL protein in targeting Lck to the plasma membrane of human T lymphocytes. *J. Exp. Med.* 205, 3201–3213.

Bodrikov, V., Pauschert, A., Kochlamazashvili, G., and Stuermer, C.A.O.O. (2017). Reggie-1 and reggie-2 (flotillins) participate in Rab11a-dependent cargo trafficking, spine synapse formation and LTP-related AMPA receptor (GluA1) surface exposure in mouse hippocampal neurons. *Exp. Neurol.* 289, 31–45.

Hülsbusch, N., Solis, G.P., Katanaev, V.L., and Stuermer, C.A.O. (2015). Reggie-1/Flotillin-2 regulates integrin trafficking and focal adhesion turnover via Rab11a. *Eur. J. Cell Biol.* 94, 531–545.

Ma, Y., Pandzic, E., Nicovich, P.R., Yamamoto, Y., Kwiatek, J., Paeon, S. V., Benda, A., Rossy, J., and Gaus, K. (2017). An intermolecular FRET sensor detects the dynamics of T cell receptor clustering. *Nat. Commun.* 8, 15100.

Reviewer #2

This is a potentially interesting paper showing a role for flotillins in t-cell activation. However, there are some technical / interpretative issues that require significant further experiments.

1. The basic endocytosis assay used in Figure 1 is not sound as it does not discriminate between TCR in invaginations of the plasma membrane - which may extend some significant distance away from the coverslip in the imaging set-up used - and TCR in vesicles that are topologically fully resolved from the plasma membrane (pinched off).

The photoactivation approach would indeed not be able to discriminate between invaginations of the plasma membrane and endocytic vesicles in a system where these vesicles would remain static after pinching off the plasma membrane. However, the strength of our approach is the ability to track the dynamics and intracellular movement of internalized proteins. In the cells and conditions described in the submitted manuscript, endocytosis of all PA-mCherry-tagged fusion proteins (TCR ζ , flotillins, clathrin) was systematically concurrent with rapid and sustained movements of the internalized vesicles. This observation would not be compatible with endocytic pits or invaginations that would not have pinched off of the plasma membrane, which would be observed as static over the timescales of our experiments. In fact, the vesicles were static only when the pinching off was prevented by dynamin or Arp2/3 inhibition (Fig. 1 e-h), providing a way to discriminate between complete endocytosis followed by intracellular trafficking and accumulation in endocytic pits.

2. Assertion that endocytosis is clathrin-independent is not sound as there is no positive control - pitstop should block tf uptake. This experiment could be better performed with one of the well characterised dom -ve mutants that clock clathrin function, such as the c-terminal domain of AP180.

We fully agree with the reviewer on the importance of performing proper positive controls regarding the effect of Pitstop2. As suggested, we have assessed the effect of Pitstop2 on transferrin uptake using Alexa488-labelled transferrin. We could not observe internalization of transferrin-Alexa488 in the presence of 6 or 12 μ M of Pitstop2, even after 30 min at 37C. Representative images of these experiments are now shown in Supplementary Fig 2d (and mentioned in the main text at line 124). This result is also in accordance with the data presented in the original manuscript in Supplementary Fig 2c, showing that cells treated with Pitstop2 form significantly less clathrin-coated vesicles than their untreated counterparts.

We have further followed the reviewer's suggestion to use the c-terminal domain of AP180 as an inhibitor of clathrin-mediated endocytosis. In accordance with the data obtained using Pitstop2, internalization of TCR ζ -PA-mCherry was not impaired in cells overexpressing c-term-AP180. These data have been added to Fig.2 (panels a and b) and are mentioned in the text at lines 128-132.

In summary:

- 1) Internalization of TCR ζ was not affected by Pitstop2, which, in the same conditions, prevented the formation of clathrin-coated vesicles and abolished transferrin internalization.
- 2) Internalization of TCR ζ was not affected by the expression of to the dominant negative c-terminal domain of AP180.
- 3) In Jurkat T cells, inhibition of Arp2/3 activity with CK666 prevented pinching off of TCR ζ - and flotillin-positive vesicles, but had no effect on clathrin-coated vesicles.

Taken together, these results allow us to confidently state that TCR ζ is not internalized through clathrin-coated vesicles in activated Jurkat T cells.

Reviewer #3

In this report the authors describe a new clathrin-independent and flotillin-dependent mechanism which controls TCR recycling to the immune synapse (IS), impacting on TCR-dependent signaling and T cell activation.

Although TCR trafficking through the endocytic compartment has strongly emerged in recent years as a central player in IS assembly and function, the pathway that orchestrates sorting at the plasma membrane, endocytosis and recycling of the TCR are only beginning to be elucidated. In this respect, the data presented in this manuscript, where a photoactivation approach has been used to follow at the single-cell level the dynamics of the TCR at the different steps of its journey from the plasma membrane to endosomes and back have significant elements of novelty. The results provide evidence that the TCR is associated with a new dynamic endocytic network marked by flotillins that regulates its nanoscale organization at the IS and promotes early and late signaling events essential to T cell activation.

Overall the experimental approach is sound and the data are solid. The work has however been entirely performed on Jurkat T cells. Although there are objective technical difficulties that preclude extending the majority of the experiments to primary T cells, the authors should attempt to validate at least the functional data (i.e. outcome of flotillin knockdown on T cell activation) on primary T cells. An endpoint such as expression of activation markers (e.g. CD69, CD25) in flotillin KO Jurkat cells should be moreover included.

We have obtained a flotillin2 knock-out mouse from the group of T Mak (Berger et al., 2012) to perform the suggested experiments on primary T cells. The lack of flotillin2 expression in splenocytes isolated from these mice resulted in a complete downregulation of flotillin1, making these mice a *de facto* flotillin double knock-out (Fig. 8a).

CD4 T cells isolated from the spleen of these mice and stimulated on surfaces coated with anti-CD3 and anti-CD28 had significantly less CD69 and CD25 surface expression after 20h and 36h of stimulation respectively (Fig. 8c and b, and lines 373-380 in the main text)

Specific points

1. *Figure 1. In this and subsequent figures the authors refer to "activated" cells to refer both to photoactivation of specific molecules (e.g. TCRzeta) and to cell activation by CD3/CD28 co-stimulation. This should be clarified in all figures.*

We apologize for the confusing labeling in figures 1 and 2. We now use "activated" for CD3/CD28 co-stimulation and "PAct." for photoactivation.

2. *Figure 1, panels g,h. Although the effects of dynasore and CK666 are clear, the authors should discuss the possible mechanisms involving dynamin and Arp2/3 that regulate the motility of the TCRzeta+ vesicles. Also, they propose that in the absence of functional dynamin or Arp2/3 the TCR remains associated with vesicles that fail to undergo fission and remain associated to the plasma membrane. This is not proven formally by the results, as the vesicles could alternatively pinch off but remain very close to the plasma membrane. The statement should be tuned down.*

The reviewer is correct, we did not formally show that TCR vesicles are still bound to the plasma membrane in dynasore and CK666 treated T cells (by, for instance, using electron microscopy). We have moderated the text as suggested (lines 103 and 112-118).

We nevertheless would like to clarify why we think that the immobility of TCR endocytic vesicles is related to an impaired pinching off of the plasma membrane rather than to ineffective transport: a) the static vesicles got brighter with time (Fig. 1g and h). This could be explained by the fact they keep accumulating TCR ζ -PAmCherry molecules coming from the photoactivated region of interest at the plasma membrane, which would be possible only if they were still connected to the plasma membrane. b) imaging of tubulin and TCR ζ vesicles as shown in Supplementary Fig. 4a suggests that these vesicles are transported along microtubules and do not rely on actin branching for their movement. c) previous studies report that flotillin endosomes rely on microtubules for their mobility rather than actin (Cornfine *et al.*, 2011; Solis *et al.*, 2013).

3. *Figure 2. In panel b the max number of TCRzeta+ vesicles in untreated cells should be plotted as a dot distribution, similar to the Pitstop2-treated cells. In panel c the distribution of clathrin appears different in the cell expressing photoactivatable TCRzeta compared to the cell showing labelled transferrin. A similar consideration applies to flotillin in panel d. The authors should comment on this apparent discrepancy. I also suggest to use cells with similar size in both panel c and d (and add a size bar)*

We have modified Figure 2b, c, and d as suggested. We apologize for the mistaken attribution of some of the colors in Figure 2c and d.

To clarify however, the distribution of clathrin is indeed different between the two images in figure 2c. This is mainly because these are two different "pools" of clathrin: 1) on cells expressing TCR ζ -PAmCherry (on the left), all the exogenous clathrin in the cells is visible, as it is fused to EGFP; 2) in cells incubated with labelled transferrin (on the right), only the

photoactivated subpopulation of clathrin PA-mCherry is visible. It is the same for flotillin in panel d. We did not observe any effect of TCR ζ -PA-mCherry expression on the distribution of clathrin or flotillin positive vesicles. Nonetheless, we have followed the reviewer's suggestion and tried to show cells with more consistent appearance and size.

4. Figure 3, panel h. The constant increase of TCRzeta+ vesicles after repetitive photoactivation should be discussed in more detail.

The constant increase of TCR ζ positive vesicles after repetitive photoactivation is now discussed in more detail in the main text (lines 200-205). This result suggests that TCR ζ -PA-mCherry molecules diffused fast enough to at least partially replete the photoactivated region of interest within the 8 sec between the photoactivation pulses. The steady number of vesicles for 450 sec following one activation and the increasing number of vesicles resulting from repetitive photoactivations indicate that, following internalization, TCR ζ is only returned slowly to the plasma membrane and remains for a significant time in endocytic compartments.

5. Figure 4. The experiments should be extended to the transferrin receptor, which the authors show to be associated with clathrin (Fig.2) and should be therefore be unaffected by flotillin KO.

Fusing the transferrin receptor with PA-mCherry unfortunately resulted in an almost total inhibition of the PA-mCherry photoactivation. Instead, we used transferrin labelled with Alexa488 to visualize clathrin-dependent endocytosis as in Fig. 2. We could not detect any difference of transferrin endocytosis between WT and flotillin1/2 KO cells in the formation of endocytic vesicles containing transferrin-Alexa488 (Supplementary Fig. 5b-d and lines 236-237 in the main text).

6. Figures 4 and 5. The authors state that flotillins1/2 KO inhibits TCR delivery to the IS (Fig.5) while not affecting TCR internalization and vesicle number, as assessed by measuring the number of TCRzeta+ vesicles 250 sec after photoactivation (Fig 4). I expect that at time points longer than 250 sec (for example 5 min, when the authors find some TCR recycling at the surface; Fig.4i), the number of TCRzeta+ vesicles would decrease due to their fusion with the plasma membrane while their number does not change or even increases slightly in flotillin KO cells (Fig.4c; also here a dot distribution should be used for WT cells similar to the KO cells). The authors should show a longer time course of vesicle number tracking.

We have performed a longer time-course as suggested by the reviewer and observed significantly more TCR ζ -PA-mCherry positive vesicles 10 min after internalization in the flotillin1/2 KO cells than in WT cells (Fig. 4 j,k). This evidence suggests that internalized TCR ζ fails to recycle properly and accumulates in intracellular compartments in the absence of flotillins. We have modified the manuscript to include this observation (lines 273-279).

The data also suggest a decrease in the number of TCR ζ -PA-mCherry positive vesicles between 1 and 10 min for the WT and an increase for the flotillin1/2 KO cells, although these changes were not significant (see Fig. R1 just below). A decrease in TCR containing vesicles in WT cells would suggest that these vesicles start fusing back to the plasma membrane.

Figure R1: Comparison of the number of endocytic vesicles detected 1 and 10 min after internalization in WT and flotillin1/2 KO cells.

7. Figure 6, panel d. The 30 ± 4 nm population of TCR ζ + clusters are clearly absent in activated flotillin1/2 KO cells. However it is not clear why the frequency of the 16 ± 2 nm cluster population is highly increased in flotillin1/2 KO cells compared to WT cells. The authors should comment on this point.

The 16 ± 2 nm population is at the limit of what our assay can detect and we do not feel confident to draw definitive conclusions from the very small changes that can – or cannot – be observed in Fig. 6d. Our interpretation is that there is no significant change in this population, which would suggest that the larger 30 ± 4 nm clusters do not result from the concatenation of the smaller 16 ± 2 nm. We have added a sentence in the manuscript to comment on this point (lines 326-330).

8. Figure 7, panels a and b. The authors describe the effects of flotillin 1/2 KO on activation-dependent T-cell signaling, using flow cytometry to measure the efficiency of conjugate formation and flow cytometry to quantify the extent of phosphorylation of key signaling molecules. I would suggest to measure some early signaling events (pTyr, P-ZAP-70, CD3 accumulation) in T cell-Raji cell conjugates to verify whether the conjugates that are formed are functional.

As suggested by the reviewer, we have stained T cell-Raji B cell conjugates for pTCR ζ and pZap70 to verify if the conjugates formed between flotillin1/2 KO cells and Raji B cells were functional. These immunofluorescence results showed a similar trend as the data obtained by flow cytometry following anti-CD3 and anti-CD28 stimulation. We consistently observed a moderate decrease of pTCR ζ and pZap70 in the flotillin1/2 KO cells when compared to the WT cells, indicating that the conjugates formed by the flotillin1/2 KO displayed impaired signaling but were at least partially functional. These data can be found in Supplementary Fig. 8 and are mentioned in lines 363-366 in the main text.

Minor points

- 1. Line 280: change TCRz with the corresponding symbol TCR ζ*
- 2. In the legend to supplementary figure 2, the authors describe a panel “d” which is not shown in the figure.*
- 3. In the legend to supplementary figure 4, the description of panel “b” is missing.*
- 4. Line 313. Supplementary figure 8 is not included in the supplemental material. I guess that the authors are referring to supplementary figure 7.*

We thank the reviewer for bringing these mistakes to our attention and have corrected the manuscript accordingly.

References:

- Berger, T., Ueda, T., Arpaia, E., Chio, I.I.C., Shirdel, E.A., Jurisica, I., Hamada, K., You-Ten, A., Haight, J., Wakeham, A., et al. (2012). Flotillin-2 deficiency leads to reduced lung metastases in a mouse breast cancer model. *Oncogene* 32, 4989–4994.
- Cornfine, S., Himmel, M., Kopp, P., El Azzouzi, K., Wiesner, C., Krüger, M., Rudel, T., and Linder, S. (2011). The kinesin KIF9 and reggie/flotillin proteins regulate matrix degradation by macrophage podosomes. *Mol. Biol. Cell* 22, 202–215.
- Solis, G.P., Hülsbusch, N., Radon, Y., Katanaev, V.L., Plattner, H., and Stuermer, C.A.O. (2013). Reggies/flotillins interact with Rab11a and SNX4 at the tubulovesicular recycling compartment and function in transferrin receptor and E-cadherin trafficking. *Mol. Biol. Cell* 24, 2689–2702.

Second revision:

Referee #1

The authors have addressed by concerns and made appropriate modification to the text. I have no further concerns.

We thank the reviewer for the positive comment and for acknowledging the modification made to the text.

Reviewer #2

The authors have addressed my previous criticisms. I point out that they have not proved that it is the recycling of TCR that is responsible for the flotillin-dependent T-cell activation phenotypes, but think that nevertheless the ms will be of interest to people working in the relevant fields.

We thank the reviewer for acknowledging that we have addressed their original concerns and that the manuscript is of interest.

We apologise for the misunderstanding regarding the new point raised by the reviewer. It was not our intention to state that we have demonstrated a direct and irrefutable link between the recycling mediated by flotillins and T cell activation. It was only our favoured hypothesis, considering **a)** published evidence showing that endocytic recycling of TCR is required for T cell activation¹⁻¹⁰ and the data presented in this manuscript showing that flotillins are required for both **b)** TCR recycling and **c)** T cell activation.

We have nevertheless tried to include further experiments supporting our hypothesis. Solely inhibiting the contribution of flotillins to recycling while preserving their other potential contributions to T cell activation proved to be challenging. We have designed an experimental protocol, adapting an optogenetic tool recently published^{11,12} to artificially aggregate and disable flotillin-positive endosomes. The percentage of Jurkat T cells positive for phosphorylated TCR in response to activation was reduced upon aggregation of flotillin positive endosomes (Supplementary figure 9 and lines 372-377), suggesting a link between flotillin-mediated recycling and T cell activation.

We are aware that this new data is not evidence that it is only the absence flotillin-mediated recycling that results in impaired T cell activation observed in flotillin knockout T cells. We have therefore amended the original text of the manuscript to avoid possible any misunderstanding regarding what we have experimentally demonstrated (lines 15, 53 and 420).

References:

1. Onnis, A., Finetti, F. & Baldari, C. T. Vesicular Trafficking to the Immune Synapse: How to Assemble Receptor-Tailored Pathways from a Basic Building Set. *Front. Immunol.* **7**, 50 (2016).

2. Soares, H., Lasserre, R. & Alcover, A. Orchestrating cytoskeleton and intracellular vesicle traffic to build functional immunological synapses. *Immunol. Rev.* **256**, 118–32 (2013).
3. Monjas, A., Alcover, A. & Alarcón, B. Engaged and Bystander T Cell Receptors Are Down-modulated by Different Endocytotic Pathways. *J. Biol. Chem.* **279**, 55376–84 (2004).
4. Barr, V. A. *et al.* T-cell antigen receptor-induced signaling complexes: internalization via a cholesterol-dependent endocytic pathway. *Traffic* **7**, 1143–62 (2006).
5. Liu, H., Rhodes, M., Wiest, D. L. & Vignali, D. A. . On the Dynamics of TCR:CD3 Complex Cell Surface Expression and Downmodulation. *Immunity* **13**, 665–675 (2000).
6. Das, V. *et al.* Activation-induced polarized recycling targets T cell antigen receptors to the immunological synapse; involvement of SNARE complexes. *Immunity* **20**, 577–88 (2004).
7. Soares, H. *et al.* Regulated vesicle fusion generates signaling nanoterritories that control T cell activation at the immunological synapse. *J. Exp. Med.* **210**, 2415–33 (2013).
8. Blas-Rus, N. *et al.* Aurora A drives early signalling and vesicle dynamics during T-cell activation. *Nat. Commun.* **7**, 11389 (2016).
9. Finetti, F. *et al.* Intraflagellar transport is required for polarized recycling of the TCR/CD3 complex to the immune synapse. *Nat Cell Biol* **11**, 1332–1339 (2009).
10. Osborne, D. G., Piotrowski, J. T., Dick, C. J., Zhang, J.-S. & Billadeau, D. D. SNX17 Affects T Cell Activation by Regulating TCR and Integrin Recycling. *J. Immunol.* **194**, (2015).
11. Nguyen, M. K. *et al.* Optogenetic oligomerization of Rab GTPases regulates intracellular membrane trafficking. *Nat. Chem. Biol.* **12**, 431–436 (2016).
12. Park, H. *et al.* Optogenetic protein clustering through fluorescent protein tagging and extension of CRY2. *Nat. Commun.* **8**, 30 (2017).

Reviewer #3

The results presented in this report provide evidence that the TCR is associated with a new dynamic endocytic network marked by flotillins that regulates its nanoscale organization at the immune synapse and promotes early and late signaling events essential to T cell activation. The data have significant elements of novelty. The experimental approach is sound and the data are solid. The authors have satisfactorily addressed all the issues raised in my previous review, adding substantial additional experimental evidence to support their findings.

We thank the reviewer for their comment and for highlighting both the novelty and the solidity of the data presented in the manuscript, as well as the substantial additional experimental evidence provided in the first revision.